

# Process-level improvements in CMIP5 models and their impact on tropical variability, Southern Ocean and monsoons

Axel Lauer[1], Colin Jones[2,3], Veronika Eyring[1], Martin Evaldsson[4], Stefan Hagemann[5], Jarmo Mäkelä[6], Gill Martin[3], Romain Roehrig[7], and Shiyu Wang[4]

[1]Deutsches Zentrum für Luft- und Raumfahrt (DLR), Institut für Physik der Atmosphäre, Oberpfaffenhofen, Germany
[2]University of Leeds, Leeds, UK
[3]Met Office Hadley Centre, Exeter, UK
[4]Swedish Meteorological and Hydrological Institute (SMHI), Norrköping, Sweden
[5]Max Planck Institute for Meteorology (MPI-M), Hamburg, Germany
[6]Finnish Meteorological Institute (FMI), Helsinki, Finland
[7]CNRM, Météo-France/CNRS, Toulouse, France

*Correspondence to:* Axel Lauer (axel.lauer@dlr.de)

**Abstract.** The performance of improved versions of the four earth system models (ESMs) CNRM, EC-Earth, HadGEM, and MPI-ESM is assessed in comparison to their predecessor versions used in Phase 5 of the Coupled Model Intercomparison Project. The earth System Model Evaluation Tool (ESMValTool) is applied to evaluate selected climate phenomena in the models against observations. This is the first systematic application of the ESMValTool to assess and document the progress made during an extensive model development and improvement project. This study focuses on the South Asian (SAM) and West African (WAM) monsoons, the coupled equatorial climate, and Southern Ocean clouds and radiation, which are known to exhibit systematic biases in present-day ESMs.

The analysis shows that the tropical precipitation in three out of four models is significantly improved. Two of three updated coupled models show an improved representation of tropical sea surface temperatures with one coupled model not exhibiting a double Inter-Tropical Convergence Zone. Simulated cloud amounts and cloud-radiation interactions are improved over the Southern Ocean. Improvements are also seen in the simulation of the SAM and WAM, although systematic biases remain in regional details and the timing of monsoon rainfall. Analysis of simulations with EC-Earth at different horizontal resolutions from T159 up to T1279 shows that the synoptic-scale variability of precipitation over the SAM and WAM regions improves with higher model resolution. The results suggest the reasonably good agreement of modeled and observed mean WAM and SAM rainfall in lower resolution models may be a result of unrealistic intensity distributions.

## 1 Introduction

Despite the progress made over the last years, global climate models (GCMs) and earth system models (ESMs) still show significant systematic biases in a number of key features of the simulated climate system compared with observations. Such systematic errors in the representation of observed climate features and their variability introduce considerable uncertainty in



model projections of future climate. Examples of such biases include the simulation of a too thin Arctic sea ice (Shu et al., 2015), systematic problems in simulating monsoon rainfall (Turner and Annamalai, 2012;Turner et al., 2011), a dry soil moisture bias in mid-latitude continental regions, an excessively shallow equatorial ocean thermocline and double-ITCZ (e.g., Li and Xie (2014)), too thick clouds in mid-latitudes (Lauer and Hamilton, 2013) and excessive downwelling solar

radiation over the Southern Ocean, accompanied by a warm bias in sea surface temperatures (SSTs) in many coupled models (Trenberth and Fasullo, 2010). This paper presents and documents the progress made in the European Commission's 7th Framework Programme (FP7) project "Earth system Model Bias Reduction and assessing Abrupt Climate change" (EMBRACE). EMBRACE specifically aimed at reducing a number of these systematic model biases by targeting improvement in the representation of selected key variables and processes in ESMs: (1) The representation of the coupled

tropical climate: (i) a cold bias in equatorial SSTs coupled with an incorrect location of the ITCZ (Lin, 2007), (ii) a poor representation of coastal and associated Ekman dynamics in the tropical oceans (de Szoeke et al., 2010) and (iii) a poor representation of the location, intensity distribution and seasonal/diurnal cycles of precipitation in monsoon regions (Kang et al., 2002). (2) Southern Ocean processes, including (i) an underestimate of reflected solar radiation at the top of the atmosphere (TOA) and an overestimate of downwelling solar radiation at the ocean surface, (ii) systematically too shallow

ocean mixed-layers, particularly in austral summer and (iii) warm SST biases across the Southern Ocean ((IPCC), 2007).
The community model evaluation and performance metrics Earth System Model Evaluation Tool (ESMValTool, Eyring et al. (2016b)) is used to evaluate a range of variables and climate processes in the models that have been improved during EMBRACE ("EMBRACE models") against observations and their CMIP5 (Coupled Model Intercomparison Project Phase 5, Taylor et al. (2012)) predecessor versions ("CMIP5 models"). The study is particularly focusing on evaluating processes

relevant to clouds and precipitation and aims at assessing the progress that has been made by model improvements introduced during the development and preparation of the models for the 6th Phase of CMIP (CMIP6, Eyring et al. (2016a)). This article is organized as follows: Sect. 2 gives an overview on the model improvements and model simulations analyzed. The improved models are then evaluated against observations and compared to the original CMIP5 versions of the models in Sect. 3, where a number of the aforementioned systematic biases are investigated. A summary of the model improvements

and outstanding biases is given in Sect. 4.

## 2 Model improvements, experiment setup and observational data

### 2.1 Model improvements

In the following, a brief summary of the main improvements of the CMIP5 models implemented during the EMBRACE project period is given. For descriptions of the individual models and details on the specific improvements the reader is

30 directed to the references listed in Table 1 and further references within these model description papers. The updated model versions evaluated here are models that are in the process of being further developed for CMIP6.

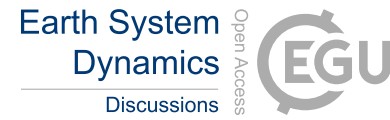

### 2.1.1 CNRM

Major changes implemented into the atmosphere component of the CNRM-CM5.1 model ARPEGE-Climat version 5 (Voldoire et al., 2013) include in particular improvements of the turbulent, convective and microphysics schemes. The new model CNRM-AM-PRE6 contains a prognostic turbulent kinetic energy (TKE) scheme (Cuxart et al., 2000) that improves the representation of the dry boundary layer while a new unified dry-shallow-deep convection scheme allows for a better transition between convective regimes (Guérémy, 2011;Piriou et al., 2007). The convective scheme solves a prognostic equation for the updraft vertical velocity and uses a convective available potential energy (CAPE) closure. It also features detailed prognostic microphysics (Lopez, 2002), which are consistent with the ones used for large-scale condensation and precipitation. Besides, dust aerosol optical properties have been updated, as well as surface albedo, leading, for instance, to an improved radiation budget in the West African monsoon region (Martin et al., 2017). CNRM-AM6-PRE6 features 91 vertical levels compared to 31 levels in the CMIP5 version.

### 2.1.2 EC-Earth

The atmosphere model of EC-Earth v2.3 (Hazeleger et al., 2013) has been upgraded from the Integrated Forecasting System (IFS) cy31r1 to IFS cy36r4 and the ocean model to the Nucleus for European Modelling of the Ocean (NEMO) 3.3.1. Major changes in the atmosphere are the new microphysics scheme with six hydrometeor classes including ice crystals and snow (Forbes et al., 2011), and the new Rapid Radiation Transfer Model (RRTM) (Jung et al., 2010). The resolution of the atmosphere model has been increased both horizontally and vertically from T159L62 to T255L91. The ocean component NEMO 3.3.1 is a major upgrade and features a moderate increase in the vertical resolution (from L42 to L46). The sea ice model was upgraded from the Louvain-la-Neuve Sea Ice Model 2 (LIM2) to LIM3 with an improved description of the sea ice rheology and physics. The option of LIM3 to take into account multiple sea ice categories was not used as the Arctic sea ice was found to be unstable in a multi-category setup. Improvements of the convection scheme were applied that were developed by the European Centre for Medium-Range Weather Forecasts (ECMWF) and resulted in a better representation of the diurnal cycle of convection (Bechtold et al., 2014).

### 2.1.3 HadGEM

Changes in the atmospheric component between HadGEM2 and HadGEM3 model families include the ENDGame dynamical core (Wood et al., 2014), the inclusion of a prognostic cloud and condensate scheme (PC2; Wilson et al. (2008)), increased convective entrainment/detrainment, a new orographic gravity wave drag (GWD) representation (Vosper et al., 2009), and numerous other changes (see Walters et al. (2011), Walters et al. (2014), Walters et al. (2017) for details). In addition, the vertical resolution has been increased and the model lid extended from 40 km to 85 km. Both of these changes require the model physical schemes to be revisited and adjusted to remove level-dependencies and, in some cases, for additional parametrizations to be included, such as the non-orographic GWD scheme (Scaife et al., 2002) to represent

momentum deposition by breaking of gravity waves in the upper stratosphere and mesosphere. The PC2 scheme is a distributed cloud parameterization that represents cloud cover and condensate changes occurring through changes to the environmental temperature and humidity as a result of the other physical parameterizations. In particular, condensate detrained by the convection scheme is handled directly by PC2, rather than being evaporated, detrained and re-condensed as in HadGEM2. Many other changes to the clouds, microphysics and convection have also been made in order to achieve a reasonable global climatology and radiative balance.

HadGEM3-GC2 (Williams et al., 2015) includes Global Atmosphere 6.0 (Walters et al., 2017), Global Ocean 5.0 (based on NEMO v3.4) with 75 vertical levels and Global Sea Ice 6.0 (see Table 1). In addition, HadGEM3-GC2 does not include earth system components such as an interactive carbon cycle, dynamic vegetation, tropospheric chemistry or ocean biogeochemistry that are present in the CMIP5 version HadGEM2-ES, but it does include interactive aerosols (with a different tuning for the dust scheme).

### 2.1.4 MPI-ESM

ECHAM6 and its land component JSBACH have undergone several further developments since the version used for CMIP5 (ECHAM6.1/JSBACH 2.0). Several bug fixes in the physical parameterizations of ECHAM6.3 assure energy conservation in the total parameterized physics. A re-calibration of the cloud processes resulted in a medium range climate sensitivity of about 3 K of the new model system. JSBACH 3.0 comprises several bug fixes, a new soil carbon model (Goll et al., 2015) and a 5-layer soil hydrology scheme (Hagemann and Stacke, 2015) replacing the previous bucket scheme.

### 2.2 Model experiments

Two kinds of model simulations have been performed, Atmosphere Model Intercomparison Project (AMIP) type simulations, i.e. atmosphere-land only with prescribed SSTs, and coupled $CO_2$ concentration-driven (historical) simulations. AMIP simulations were performed with all four improved models (EC-Earth3, HadGEM3-GA6 (denoted HadGEM3-A hereafter), CNRM-AM-PRE6, MPI-ESM), the three models EC-Earth3, HadGEM3-GC2 and MPIESM_1_1 were used to perform coupled simulations. For both types of simulations the CMIP5 protocol was followed (Taylor et al., 2012). The model experiments analyzed are summarized in Table 2. The main focus of this study will be on the coupled simulations, as these model configurations are particularly relevant to projecting future climate change.

### 2.3 Observational data

The observational and reanalysis data used for the model evaluation are summarized per data set in Table 3 and the variable definitions are given in Table 4.



## 3 Comparison of improved models with predecessor versions and with observations

### 3.1 Near-surface temperature and precipitation

Near-surface air temperature and precipitation are controlled by a large number of interacting processes making it challenging to understand and improve model biases in these quantities as model errors can partly compensate each other. The two variables, however, are frequently analyzed in atmospheric models and can provide an overview and a starting point for further analysis.

### 3.1.1 Near-surface air temperature

Figure 1 shows the bias in 20-year annual mean near-surface temperatures averaged over the years 1986-2005 from the CMIP5 and EMBRACE models compared with the observationally constrained reanalysis of the global atmosphere and surface conditions ERA-Interim (Dee et al., 2011). All data have been interpolated to a common 1° x 1° grid using a bilinear interpolation scheme. The mean near-surface temperature from the individual models agrees with the ERA-Interim reanalysis mostly within ±3 °C. Larger biases can be seen in regions with sharp gradients in temperature, for example in areas with high topography such as the Himalayas and the sea ice edge in the Southern Ocean.

In the AMIP simulations (left two columns in Figure 1), the model MPI-ESM shows only very modest changes in the simulated mean near-surface temperature bias, whereas particularly EC-Earth3 and CNRM-AM-PRE6 show considerable improvements compared with their CMIP5 versions over North America. The cold biases over large parts of Antarctica found in CNRM-CM5 are also reduced in the EMBRACE version of the model, possibly related to improvements in the turbulence scheme and the increased vertical resolution in the lower troposphere. In contrast, the warm bias over Central Africa in CNRM-AM-PRE6 and HadGEM3-A is worse compared with their CMIP5 counterparts and might be partly related to reduced (convective) precipitation in this region (see also Figure 2) in the EMBRACE versions of the models. In the HadGEM3-A model, the increase in the near-surface temperature bias over India seems to be related to less summer monsoon rainfall (see also Sect. 3.1.2).

In the concentration-driven historical coupled simulations (right two columns in Figure 1), EC-Earth3 shows a bias reduction over many parts of the continents as well as over tropical and subtropical oceans, in particular over the Southern Ocean, Central Africa and Northwest America. Despite these bias reductions, the globally averaged mean bias remains similar at about -1.1 °C. This is a consequence of reductions in the warm bias e.g. in the Southern Ocean leading to less error compensation of negative biases in other regions. While the CMIP5 version HadGEM2-ES shows a globally averaged negative bias of about -0.4 °C in near-surface temperature, HadGEM3-GC2-N96 has a positive global average bias (~0.7 °C). This is particularly caused by larger positive biases over most parts of the southern hemisphere ocean as well as over the tropical areas of Africa and South America in HadGEM3-GC2-N96. In these regions, the near-surface temperature biases in the EMBRACE version are up to 2 °C larger than in the predecessor version. Williams et al. (2015) comment that, while both models have a large downwards surface flux bias over the Southern Ocean, the larger coupled SST (and upper ocean





heat content) biases appear to be related to changes to both the lateral and vertical ocean heat transports associated with the change in ocean model and ocean resolution. The HadGEM3-GC2 errors also include a contribution associated with too shallow Southern Ocean summer mixed layers. Model biases in HadGEM3-GC2-N96 are, however, reduced compared with the CMIP5 version particularly over the American Arctic with bias reductions of about 1 °C. The MPI-ESM model shows

only rather small changes in the simulated annual mean surface temperature between the CMIP5 and EMBRACE version. Similarly to the HadGEM3-GC2-N96 model, the warm bias over Southern Ocean is slightly worse in the EMBRACE simulation than in the CMIP5 simulation.

In the AMIP simulations, biases in the near-surface temperature climatology from the EMBRACE models are particularly reduced in mid-latitudes such as over North America but are increased in some models over many parts of the tropical

continents. In most of the analyzed models, a warm bias over Central Africa and northern South America is still present in the EMBRACE simulations. In the coupled simulations, large biases are still present in the Southern Ocean, in particular along the coast of Antarctica.

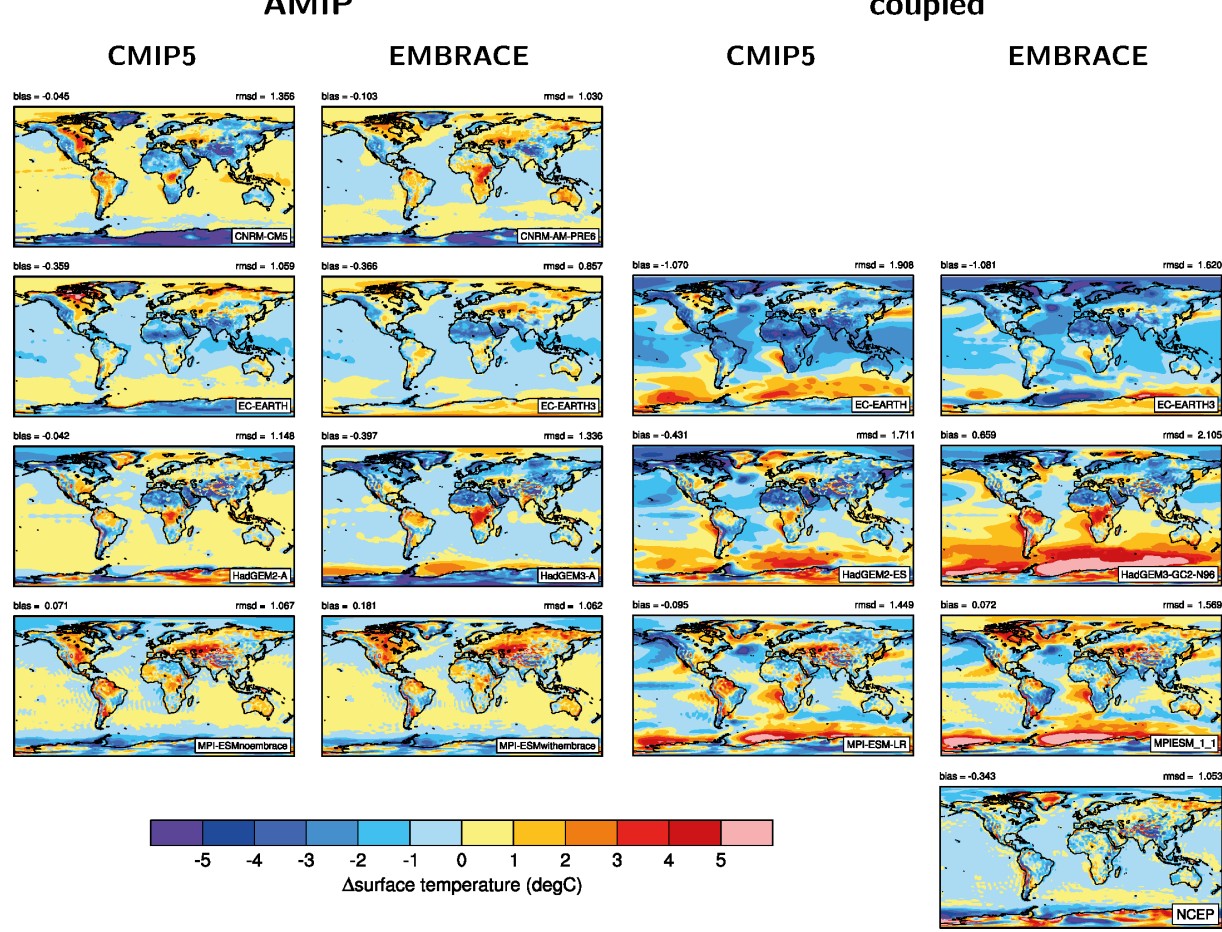

**Figure 1. Bias in 20-year annual mean near-surface air temperature for the period 1986-2005 (MPI AMIP models: 1980-1999).**
**Shown are the differences between the 20-year climatology from ERA-Interim and from left to right (1) the AMIP simulations**



**from the CMIP5 models (2) the corresponding EMBRACE models (3) the coupled historical simulations from the CMIP5 models, and (4) the corresponding EMBRACE models. As a second reference data set, data from the NCEP reanalysis are shown in the lowest rightmost panel.**

### 3.1.2 Total precipitation

Biases commonly found in the simulated mean precipitation from CMIP5 models include too little precipitation along the equator in the western Pacific associated with ocean-atmosphere feedbacks (Collins et al., 2010) and too high precipitation amounts in the tropics south of the equator related to an unrealistic double-ITCZ in many models, particularly in the Pacific (Oueslati and Bellon, 2015).

Figure 2 shows the biases in annual mean precipitation averaged over the 20-year period 1986-2005 from the CMIP5 and EMBRACE simulations compared with data from the Global Precipitation Climatology Project (GPCP, Adler et al. (2003)). The model data have been interpolated to the 2.5° x 2.5° grid of the GPCP observations using a bilinear interpolation scheme. In contrast to the AMIP simulations with the MPI-ESM model showing no large changes in the amplitude and geographical distribution of the precipitation bias between the CMIP5 version (MPI-ESMnoembrace) and the EMBRACE version (MPI-ESMwithembrace), the EMBRACE models CNRM-AM-PRE6, EC-Earth3, and HadGEM3-A show considerable reductions in the precipitation biases compared with their CMIP5 versions. The CNRM-AM-PRE6 AMIP simulation shows a considerable reduction in the wet bias over large parts of the tropical ocean by up to 2 mm day$^{-1}$ but a slightly worse dry bias in the tropical regions of South America and Africa compared with the CMIP5 simulation from CNRM-CM5. EC-Earth3 also shows a reduction in the wet bias over most parts of the tropical oceans by about 1 mm day$^{-1}$ and in addition a small reduction in the dry bias over the tropical parts of South America and Africa in comparison to EC-Earth. While the pattern of precipitation biases in HadGEM3-A is similar to that in HadGEM2-A, the magnitude of the bias is reduced in many regions, particularly over the tropical Indian Ocean and West Pacific.

In the coupled model simulations (rightmost two columns in Figure 2), EC-Earth3 shows a similar reduction, compared with EC-Earth, in the dry bias over northern South America and in the wet bias over the tropical Atlantic, to that seen in the AMIP configuration. The differences between the CMIP5 and the EMBRACE simulation from EC-Earth in most other regions are rather small. The coupled simulations with the HadGEM and the MPI-ESM model perform quite similarly and do not show large differences between the EMBRACE (HadGEM3-GC2-N96: global average RMSD = 1.22 mm day$^{-1}$, MPIESM_1_1: RMSD = 1.38 mm day$^{-1}$) and the CMIP5 versions (HadGEM2-ES: RMSD = 1.25 mm day$^{-1}$, MPI-ESM-LR: RMSD = 1.48 mm day$^{-1}$) of the models. On average, the coupled EMBRACE simulation with MPIESM_1_1 results in slightly drier conditions than the one with the CMIP5 model MPI-ESM-LR.

It is noteworthy that the bias reduction in precipitation over tropical oceans with the EMBRACE models is smaller in the coupled experiments than in the atmosphere-only simulations. This is partly due to compensation between precipitation and SST biases in coupled models (e.g., Levine and Turner (2012), Vanniere et al. (2014)). Quantitative assessments are, however, not possible as the model setups of the coupled simulations analyzed here do not match exactly the ones used for the AMIP simulations.





The tropical precipitation in three out of four EMBRACE models analyzed is significantly improved, which can be partly attributed to improved convective precipitation in the models as well as other improvements in the atmospheric components of the model (e.g., EC-Earth). The wet biases in these regions in the CMIP5 simulations have been reduced by up to 1-2 mm day$^{-1}$.

5  In the following sections, more regional or process-specific climate phenomena known to exhibit systematic errors in present-day GCMs are evaluated. The following subsections cover: (i) the South Asian and West African monsoons, (ii) coupled equatorial oceanic climate, and (iii) Southern Ocean clouds and radiation.

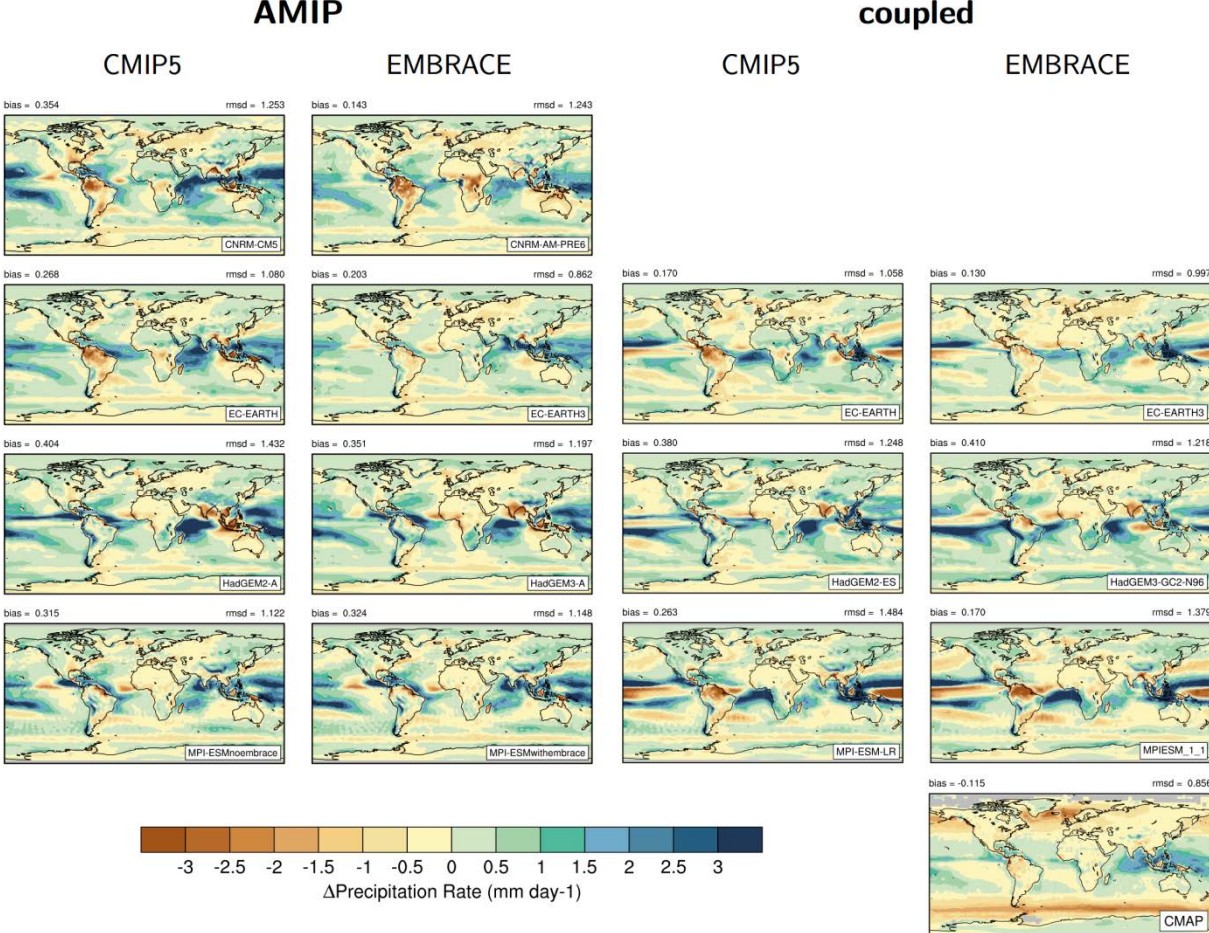

**Figure 2. Bias in annual mean precipitation rate (mm day$^{-1}$) for the 20-year period 1986-2005 (MPI AMIP models: 1980-1999) as**
10  **the difference between the Global Precipitation Climatology Project and from left to right (1) the AMIP simulations from the CMIP5 models (2) the corresponding EMBRACE models (3) the coupled historical simulations from the CMIP5 models, and (4) the corresponding EMBRACE models. Data from CMIP are shown as a second reference data set in the lowermost rightmost panel.**



### 3.2 Monsoon

### 3.2.1 South Asian monsoon

The South Asian monsoon (referred to as the SAM hereafter) provides over 1 billion people with their primary source of water (Turner and Annamalai, 2012). Accurate estimates of possible future changes in the SAM are therefore crucial for long

term planning in the region (Menon et al., 2013).

The SAM has two distinct seasonal components. The winter monsoon is dominated by a planetary-scale circulation linked to the Siberian anticyclone and Aleutian low centered over the North Pacific. These features induce northerly flow across South Asia from November to March with minimal amounts of precipitation (Chang et al., 2006). The summer monsoon starts in April, with the onset of rain over southern India and Myanmar generally occurring in early June and propagating northwest,

reaching northern India by mid-July. The monsoon rainy season extends to the end of September before reverting back to winter monsoon conditions by November (Chang et al., 2006). Due to the importance of ocean-atmosphere interactions in the evolution of the SAM and because we are primarily interested in evaluating model configurations that can be used for making future projections, here we analyze the ability of the coupled EMBRACE models to represent the main features of the summer SAM. Figure 3 shows seasonal mean (June to September, JJAS) precipitation from the coupled models and the

differences relative to the satellite product TRMM 3BV43 (Huffman et al., 2007). Figure 4 and Figure 5 show near-surface temperature and 850hPa zonal wind speed compared to data from the ERA-Interim reanalysis. Also shown are the alternative observation-based datasets GPCP-SG (precipitation) and the NCEP reanalysis (Kanamitsu et al., 2002) (near-surface temperature and zonal wind speed).



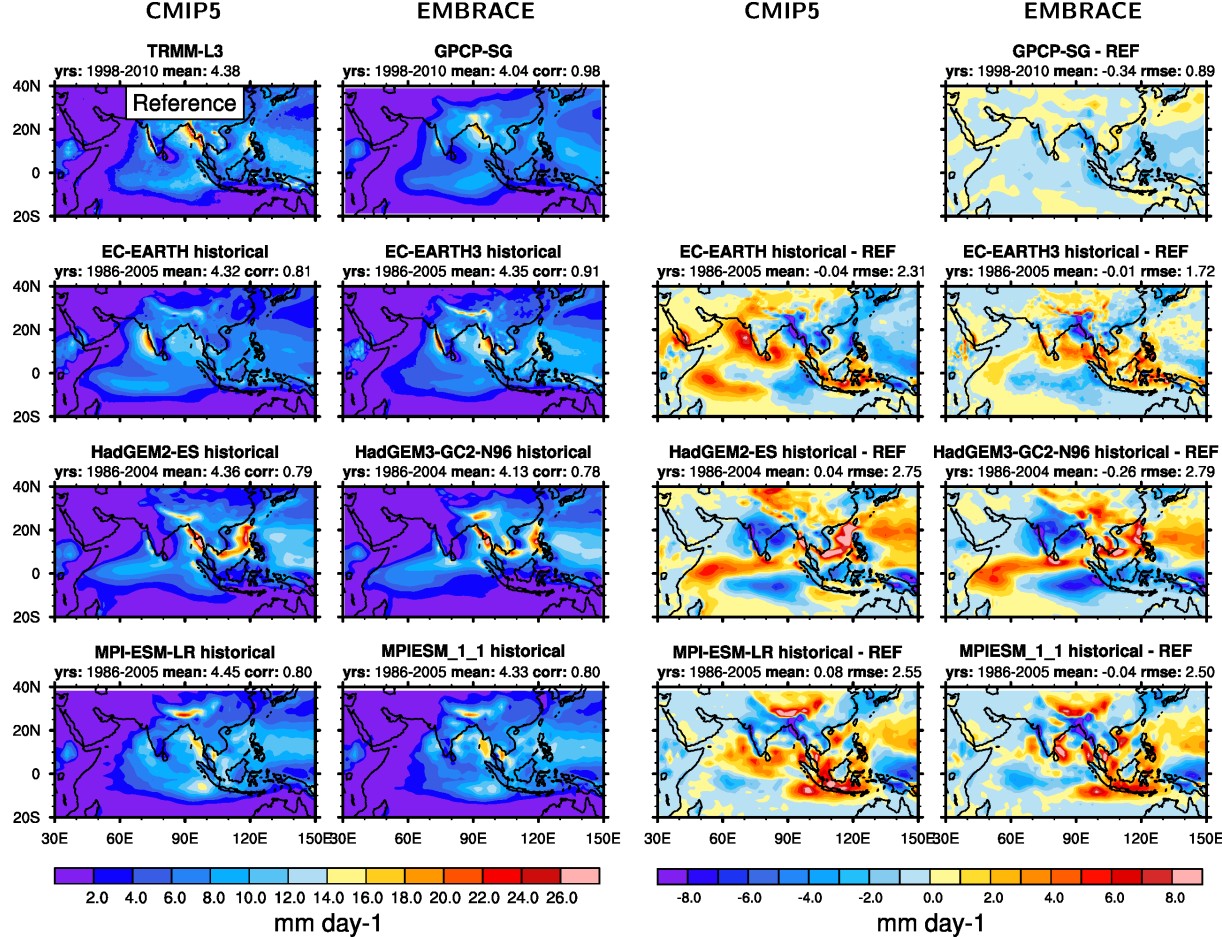

**Figure 3.** (leftmost two columns) seasonal mean precipitation for JJAS from observations (TRMM, GPCP) and the coupled simulations. (rightmost two columns) differences relative to TRMM. Columns 1 and 3 show the original CMIP5 model versions, columns 2 and 4 the EMBRACE-updated models.

In the observations, a precipitation maximum is seen on the west coast of India, with a relative minimum on the lee side of the Western Ghats. Further maxima are seen along the coast of Myanmar and Laos (east coast of the Bay of Bengal) and along the foothills of the Himalayas. A broad region of precipitation is also evident in central and north-east India. EC-Earth and MPI-ESM capture these primary rainfall features with varying degrees of accuracy. EC-Earth overestimates rainfall over the ocean adjacent to the Western Ghats and over the Bay of Bengal. Further east, over Myanmar and Laos, precipitation is

underestimated. The positive precipitation bias over the ocean is clearly improved in EC-Earth3. Both MPI-ESM versions underestimate rainfall over the Indian subcontinent, with particular negative biases associated with the Western Ghats mountains and the foothills of the Himalayas, likely caused by the low resolution of MPI-ESM. There is little difference between the two MPI-ESM configurations. The major precipitation biases are also largely unchanged between the two HadGEM models, with significant underestimate of precipitation across India and a secondary negative bias south of India

along the equator. The process of irrigation that is missing in current GCMs might contribute to the dry bias over Northern




India. Saeed et al. (2009) found that temperature biases caused by a too strong differential heating between land and sea if no irrigation is considered can lead to unrealistic simulations of the SAM circulation and associated rainfall in climate models. The HadGEM models (particularly HadGEM3-GC2) show a large warm bias in 2m temperature over the Indian land mass (Figure 4). This error is linked to excess downwelling surface shortwave radiation (of up to 60 Wm$^{-2}$ in the JJAS mean) due

5 to a lack of optically thick clouds over India. The lack of simulated rainfall exacerbates this problem further, leading to a dry land-surface bias, reduced surface evaporative cooling and increased surface sensible heat flux. The converse is seen in both EC-Earth coupled simulations, with a cold bias of ~5 °C over India linked to an underestimate of downwelling solar radiation of ~40 Wm$^{-2}$. The domain averaged cold bias in the EMBRACE simulation with EC-Earth is, however, considerably reduced from -2.1 °C in the CMIP5 version of the model to -1.3 °C in the EMBRACE version.

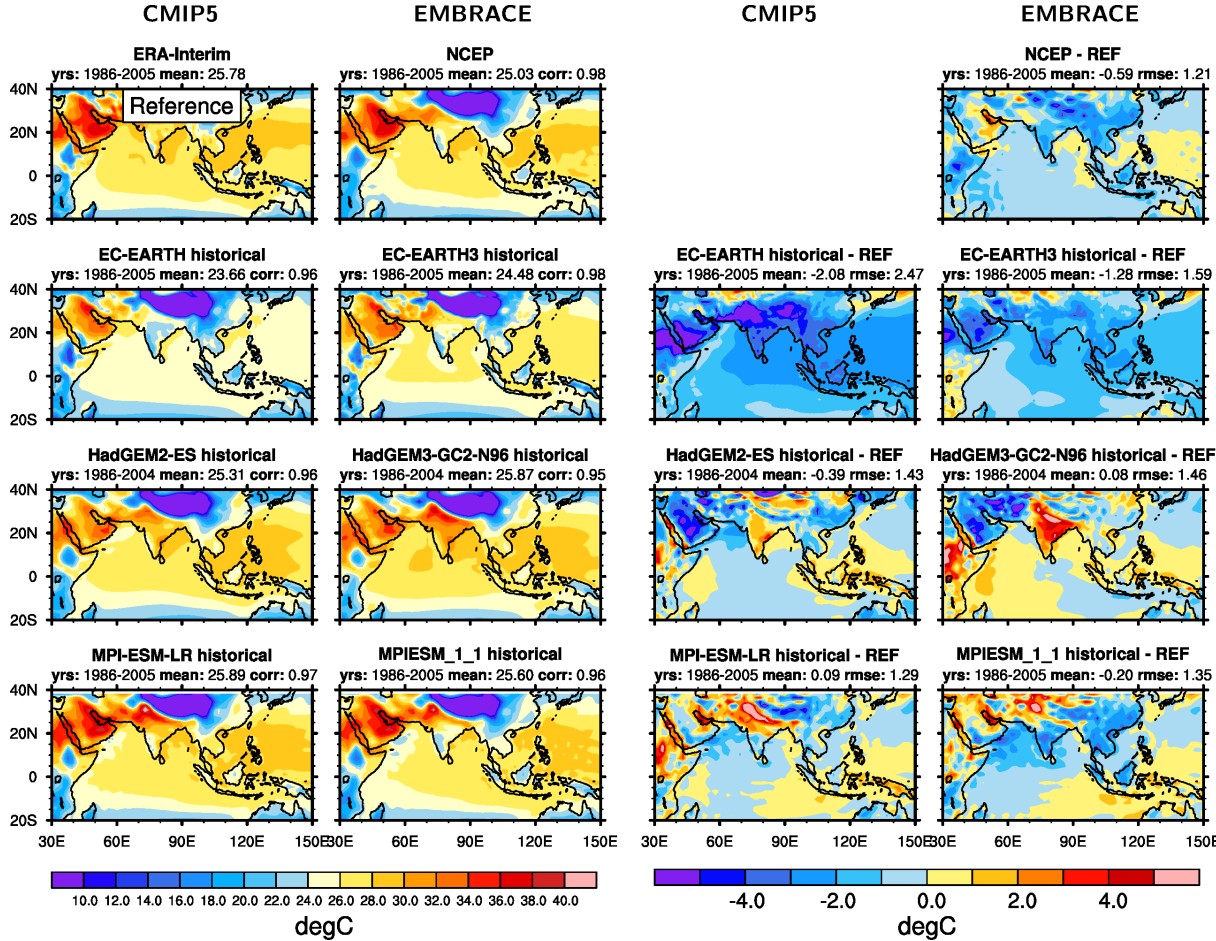

**Figure 4. (leftmost two columns) seasonal mean 2m temperature for JJAS from reanalysis data (ERA-Interim), the CRU dataset and the coupled simulations. (rightmost two columns) differences relative to ERA-Interim. Columns 1 and 3 show the original CMIP5 model versions, columns 2 and 4 the EMBRACE-updated models.**

All of the models represent the cross-equatorial low-level jet and acceleration of the westerly monsoon flow across the

15 Arabian Sea (Figure 5), though the strength of the jet core and the eastward extension of the westerlies towards the





Philippines varies between models. Positive biases in 850hPa wind speed are reduced in the HadGEM3-GC2-N96 model and are replaced by a negative bias over the Arabian Sea. In contrast, the largely negative biases in EC-Earth are replaced by a positive bias in EC-Earth3. Both EC-Earth configurations, and to a lesser extent the MPI-ESM models and HadGEM3-ES, have a cold SST bias across the western Indian Ocean and Arabian Sea (as seen from the biases in 2m temperature in Figure

4). For a given low-level wind speed a cold bias in Arabian Sea SSTs will act to decrease surface ocean evaporation (relative to the correct SST) and thus reduce the atmospheric moisture flux into India and consequently precipitation, while a cold bias in the equatorial Indian Ocean contributes to the meridional temperature gradient and thereby enhances the monsoon flow (Levine and Turner, 2012;Levine et al., 2013).

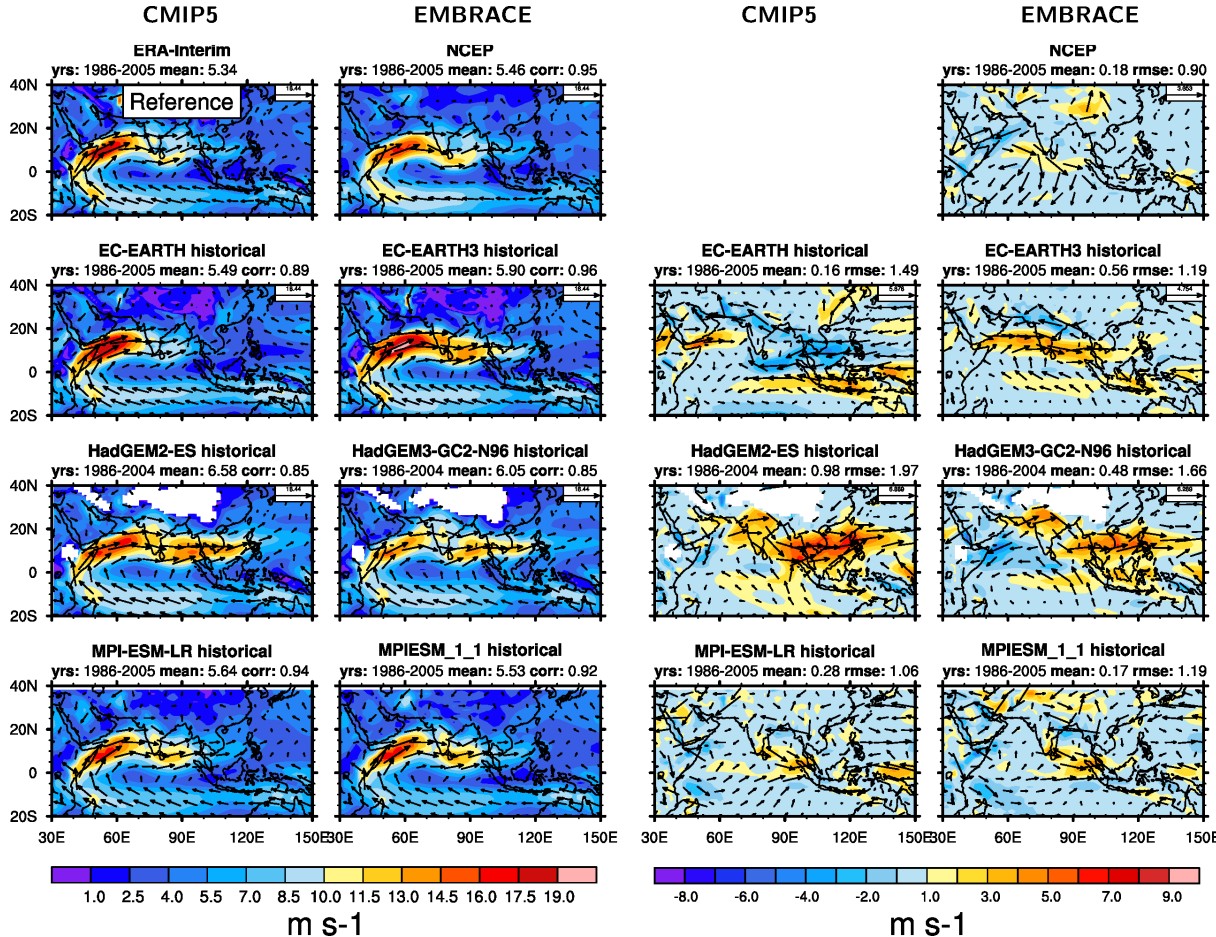

**Figure 5. (leftmost two columns) seasonal mean zonal wind speed at 850 hPa for JJAS from reanalysis data (ERA-Interim, NCEP) and the coupled simulations. (rightmost two columns) differences relative to ERA-Interim. Columns 1 and 3 show the original CMIP5 model versions, columns 2 and 4 the EMBRACE-updated models.**

Figure 6 summarizes the annual cycle of SAM sampling both precipitation and dynamical measures. Figure 6a shows the mean annual cycle of precipitation spatially averaged over 5° to 30° N and 65° to 95° W. EC-Earth overestimates both the

duration of the monsoon rainy season and the mean rainfall intensity during the peak monsoon. Both these biases are





improved in EC-Earth3. There is little difference between the two MPI-ESM models, which at this spatial scale exhibit an accurate simulation of monsoon rainfall. The two HadGEM configurations significantly underestimate rainfall, with biases particularly large in the early monsoon period (May to July). Through bias compensation the multi-model mean provides the most accurate mean annual cycle. Figure 6b shows the annual cycle of the Webster and Yang (Webster and Yang (1992), hereafter WY) dynamical monsoon index and Figure 6c the Goswami index (Goswami et al. (1999), hereafter GM). The WY index is based on vertical shear of the tropospheric zonal wind speed ($u_{850\,hpa} - u_{200\,hpa}$) averaged over 40°-110° E and 0°-20° N, while the GM index is a measure of vertical shear in the meridional wind speed ($v_{850\,hpa} - v_{200\,hpa}$) averaged over 70°-110° E and 10°-30° N. Both capture the interplay between large-scale dynamics and atmospheric diabatic heating over the Indian region. The WY index is a measure of the large-scale south-westerly monsoon circulation, while the GM index is a measure of the Hadley circulation intensity and meridional propagation. All models exhibit significantly more accuracy in simulating these dynamical measures compared to SAM precipitation, particularly the WY index, although part of this improved performance stems from error compensation between lower tropospheric (850 hPa) and upper troposphere (200 hPa) wind speed biases (not shown).

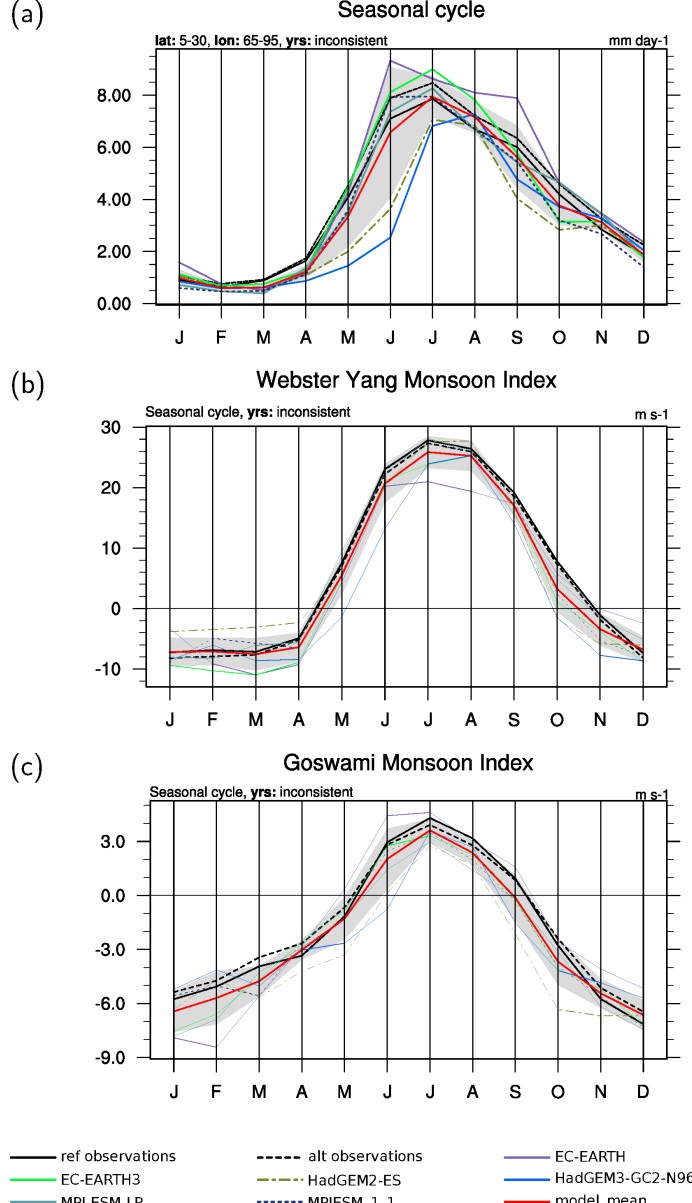

**Figure 6. Mean annual cycle plots: (a) precipitation spatially averaged over 5°-30° N, 65°-95° W, (b) Webster and Yang Monsoon Index, (c) Goswami Monsoon Index. Shown are the coupled (historical) simulations. The reference observations (a) TRMM-L3, (b, c) ERA-Interim are shown as solid black lines, the alternative observations (a) GPCP-SG, (b, c) NCEP are shown as black dashed lines.**

5  All of the EMBRACE coupled models exhibit significant biases with respect to monsoon precipitation. Only EC-Earth3 showed a measureable improvement over its CMIP5 predecessor. Most of the models appear to capture the large-scale dynamical evolution of the SAM, but fail to capture the associated evolution of precipitation, particularly the sub-continental distribution of rainfall, although at the scale of "all India" the MPI-ESM models do capture the annual cycle quite well. Models continue to have severe problems capturing the subtle interactions between deep convection, cloud-radiation



processes, precipitation, and surface evaporation and the associated interplay between diabatic heating over land and the large-scale monsoon circulation and associated oceanic evaporation.

### 3.2.2 West African monsoon

West Africa also experiences a summer monsoon from May to October (Nicholson and Grist, 2003) with rains starting in
May at the Guinea coast and propagating northward to the Sahel region (~15° N) by mid-July (Sultan and Janicot, 2003). Failures in the west African monsoon (hereafter WAM), or lack of northward propagation into the Sahel can have devastating consequences for the population of this region, as evidenced by the extensive famines during the 1970's and 1980's linked to decadal variability in WAM rainfall (Held et al., 2005;Nicholson et al., 2000). As with the SAM, the WAM also results from the seasonal development of a low-level thermal gradient between the tropical ocean and the Sahara
(Caniaux et al., 2011;Lavaysse et al., 2009). This monsoon circulation and the associated low-level moisture flow interact with westward propagating, synoptic-scale, African Easterly Waves (AEWs, (Poan et al., 2013;Poan et al., 2015)) that grow on the southern and northern flanks of the African Easterly Jet (Thorncroft and Hoskins, 1994b, a) (hereafter AEJ). This interaction between AEWs and the monsoon moisture flux supports the development of organized mesoscale convective systems (MCS) embedded within the AEWs. These MCS deliver the majority of rainfall over West Africa (Fink and Reiner,
2003;Mathon et al., 2002). Such interaction across scales (convective-meso-synoptic-planetary scales) is at the heart of WAM dynamics and is a challenge for GCMs, which prevents them to accurately simulate this system, including both natural and forced modes of variability (Biasutti, 2013;Roehrig et al., 2013;Ruti and Dell'Aquila, 2010).

Figure 7 and Figure 8 show absolute values of JJAS mean precipitation and 2m temperature as well as their biases over West Africa compared with observations from TRMM and ERA-Interim reanalysis data, respectively. Differences between
TRMM and GPCP, for precipitation, and ERA-Interim versus Climatic Research Unit (CRU, Harris et al. (2014)) data, for 2m temperature, give an estimate of observational uncertainty in the region. The WAM is marked by a maximum in precipitation stretching from the Atlantic coast across to the Darfur mountains in Sudan over a latitude band ~5° N to 15° N. Directly north of the precipitation maximum, near surface temperatures increase rapidly over the Sahara. Surface warming induces a deep near-surface low-pressure system (the Saharan heat low, Lavaysse et al. (2009)) that one of is the main
drivers of the low-level south-westerly flow into West Africa.



**Figure 7. (leftmost two columns) seasonal mean precipitation for JJAS from observations (TRMM, GPCP) and the coupled simulations. (rightmost two columns) differences relative to TRMM. Columns 1 and 3 show the original CMIP5 model versions, columns 2 and 4 the EMBRACE-updated models.**

All the coupled models exhibit a positive precipitation bias over the Gulf of Guinea. This error is reduced when the models are run with prescribed SSTs (not shown). Such precipitation errors are associated with a warm bias in all three models' SST fields off the coast of Namibia and Angola (evident in the 2m temperatures, Figure 8). The warm SST bias, in combination with the predominant southerly low-level atmospheric flow into West Africa (Figure 9) drives a large (and excessive) low-level moisture convergence into the Guinea coast region and is arguably the main cause of the precipitation bias. Positive SST biases in this region are common to coupled GCMs (Toniazzo and Woolnough, 2014), and are thought to arise from a combination of poorly resolved coastal ocean dynamics (Wahl et al., 2011;Xu et al., 2014) and atmospheric wind forcing (Richter and Xie, 2008;Voldoire et al., 2014) and poor simulation of marine stratocumulus clouds (Huang et al., 2007). EC-Earth3 has somewhat improved SST biases in this region compared to its CMIP5 version which may partly explain the reduced rainfall bias off the Guinea coast.





**Figure 8. (leftmost two columns) seasonal mean 2m temperature for JJAS from reanalysis data (ERA-Interim), the CRU dataset and the coupled simulations. (rightmost two columns) differences relative to ERA-Interim. Columns 1 and 3 show the original CMIP5 model versions, columns 2 and 4 the EMBRACE-updated models.**

5    Figure 9 shows the 925hPa wind velocity over West Africa. Strong south-westerly flow is evident from the Gulf of Guinea into West Africa. EC-Earth and both MPI-ESM models have a large westerly (zonal) bias in the low-level flow, suggestive of convergence driven by convective heating of the atmosphere over the Gulf of Guinea. These three models also show the largest positive bias in precipitation in this region. This bias is particularly marked for the EMBRACE version of the MPI-ESM model. EC-Earth3 has an improved low-level wind structure compared to EC-Earth, likely due to a combination of

10   improved SST off Angola and reduced convection over the Gulf of Guinea and a reduction of the cold bias in 2m temperatures (Figure 8). Both HadGEM models indicate southwesterly flow into West Africa but a negative (northerly) wind bias north of ~15° N, indicative of the WAM not penetrating sufficiently far north through the summer season.



**Figure 9. (leftmost two columns) seasonal mean wind speed at 925 hPa for JJAS from reanalysis data (ERA-Interim and NCEP) and the coupled simulations. (rightmost two columns) differences relative to ERA-Interim. Columns 1 and 3 show the original CMIP5 model versions, columns 2 and 4 the EMBRACE-updated models.**

5   Latitudinal cross-sections of precipitation, 2m temperature and key radiation variables averaged from 10° W to 10° E for the JJAS season are shown in Figure 10. A maximum in 2m temperatures (Figure 10a) coincides with a minimum in sea level pressure (Figure 10b) associated with the Saharan heat low (around 22° N). While there is some discrepancy between the simulated 2m temperature and the two observationally based data sets (ERA-Interim and CRU) all models capture the sharp increase in temperature around 15° N although maximum temperatures over the Sahara can vary by 5 °C across models.

10  Most models also capture the location and intensity of the Saharan heat low fairly well. Surprisingly, the warmest model over the Sahara (MPI-ESM) has the weakest low-pressure minimum and HadGEM3-GC2, with one of the deepest low pressures, has relatively cool temperatures over the Sahara. Possibly more significant, the location of the low-pressure minimum in HadGEM3-GC2 is displaced ~500 km south of the observed minimum. A key driver of high Saharan surface temperatures is surface absorption of solar radiation. Figure 10c shows downwelling surface solar radiation (SWD) with



CERES-EBAF satellite derived estimates as an observationally-based reference (Loeb et al., 2009). A relative minimum in SWD around 10° N coincides with the main band of precipitation (Figure 10g) and associated optically thick clouds. Further north SWD increases to 300 Wm$^{-2}$ at 25° N. Model SWD shows a wide spread over the Sahara ranging from 280 Wm$^{-2}$ in MPI-ESM to 330 Wm$^{-2}$ in the two HadGEM simulations. While the HadGEM models have the highest incoming SWD values over the Sahara they simulate one of the coldest Saharan 2m temperatures. This discrepancy most likely arises from a positive surface albedo bias over the Sahara in the HadGEM models. The probable cause of the variable SWD across models is either, erroneous optically thin ice clouds and/or a poor representation of Saharan dust and other aerosols and their interaction with solar radiation. MPI-ESM and EC-Earth3 have relatively accurate representation of both SWD and 2m temperature over the Sahara. Surface temperatures are significantly improved moving from EC-Earth to EC-Earth3. Downwelling surface longwave radiation (LWD, Figure 10d) is significantly underestimated by all models over the Sahara except for the MPI-ESM models.

Figure 10e shows a cross-section of the shortwave (SW) cloud radiative effect (CRE) for JJAS. Negative SW CRE indicates clouds reduce the amount of SW radiation absorbed by the atmosphere-surface system relative to an equivalent clear-sky atmosphere (i.e. increased SW reflection). This is clearly visible around 10° N where CERES indicates a reduction in absorbed SW of -90 Wm$^{-2}$ due to clouds. Cloud effects drop to -10 Wm$^{-2}$ over the Sahara. Both HadGEM models simulate SW CRE over the Sahara close to 0 Wm$^{-2}$, indicative of zero cloud cover. This may partly explain the high bias in SWD seen in HadGEM. The longwave cloud radiative effect (LW CRE) is shown in Figure 10f, with positive values indicating clouds reduce the amount of outgoing longwave radiation (OLR) relative to a clear-sky equivalent atmosphere (i.e. more terrestrial-emitted radiation trapped in the atmosphere). The precipitation-cloud maximum at 10° N is delineated by a maximum in LW CRE of 40 Wm$^{-2}$. The majority of models underestimate LW CRE compared to CERES particularly in the latitude band 10°-20° N. In this band most models also underestimate the negative cloud radiative effect SW CRE indicating model clouds in this band are optically too thin, consistent with an underestimate of rainfall in this band in most models.

Zonally averaged precipitation between 10° W and 10° E from TRMM-3B43 and GPCP-1DD show relatively good agreement (Figure 10g). The majority of models fail to represent the rapid increase in precipitation between 0° to 5° N close to the Guinea coast due to excessive precipitation over the ocean. Most models represent the second maximum in precipitation around 12° N, linked to AEWs on the southern flank of the African Easterly Jet. HadGEM, EC-Earth and MPI-ESM are all somewhat deficient in rainfall, particularly in the northern maxima region, consistent with the cloud-radiation errors discussed above. There is no clear improvement in precipitation between the CMIP5 models and the EMBRACE-improved models.



**Figure 10.** 10° W-10° E zonal average, JJAS mean values as a function of latitude (x-axis) for (a) 2m temperature, (b) sea level pressure, (c) surface downwelling solar radiation, (d) surface downwelling longwave radiation, (e) shortwave cloud radiative forcing, (f) longwave cloud radiative forcing, and (g) precipitation. Model results are for the coupled simulations. The reference observations (a, b) ERA-Interim, (c, d, e, f) CERES-EBAF, (g) TRMM-L3 are shown as solid black lines, the alternative observations (a) CRU, (f) SRB, (g) GPCP-SG are black dashed lines.

In addition to simulating seasonal mean statistics of the WAM and SAM, it is important models also represent the underlying weather variability that makes up the seasonal mean precipitation. Any future changes in intra-seasonal precipitation variability will likely have as big an impact on societies in the two regions as changes in seasonal mean monsoon rainfall.



The 3-10 day band-pass filtered variance of precipitation (Figure 11) emphasizes the dominant timescale of precipitation variability over West Africa. This variability is associated with westward propagating AEWs and MCS embedded within these waves. Both TRMM and GPCP show significant variance in precipitation at these timescales, stretching from the Darfur mountains west across the Sahel region, with maximum values westward from ~0° E to the Atlantic coast, coincident

5  with the southern flank of the AEJ.

EC-Earth appears to capture the northern band of precipitation variability quite well, although this is degraded in EC-Earth3 west of the 0° meridian. Both the HadGEM and MPI-ESM models fail to capture sufficient precipitation variability at these timescales over land, with significant variability only occurring over the tropical ocean regions. Such findings emphasize the need for an improved representation of wave-precipitation interactions in all coupled models before they can provide robust

10  estimates of changes in intra-seasonal rainfall over this region.





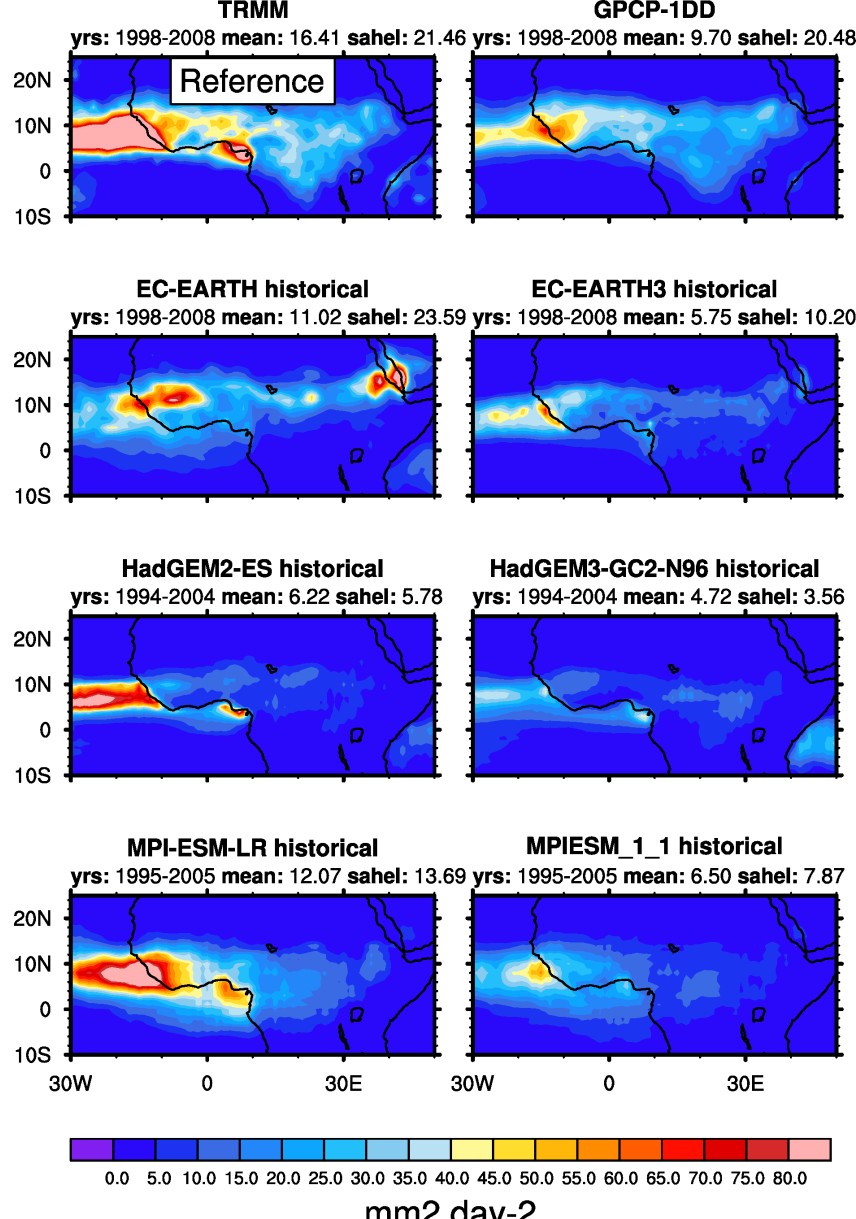

**Figure 11. JJAS average 3-10 day band-pass filtered precipitation variance (mm$^2$ day$^{-2}$) calculated from 11 years of daily precipitation fields as indicated above the panels. The top two panels show observations from TRMM (left) and GPCP-1DD (right). Shown are the coupled simulations from (left) the CMIP5 and (right) the EMBRACE-updated models.**

5    Higher model resolution is generally considered an important route for improving weather timescale variability in climate models (Jung et al., 2012;Roberts et al., 2015). In the following section we present analysis of EC-Earth simulations run with prescribed SSTs (AMIP mode) sampling horizontal resolutions from T159 (125 km) to T1279 (16 km). In this analysis we focus on the potential benefit increased atmospheric model resolution brings to simulating synoptic (weather) timescale





precipitation variability over both the WAM and SAM regions. Presently these findings are for one EMBRACE model only, but are likely pertinent to model development priorities across modeling groups.

### 3.2.3 Representing synoptic time scale precipitation variability in monsoon systems: The role of increased model resolution

While an accurate representation of the mean monsoon climatology, in particular the annual cycle, is a fundamental requirement of GCMs, rainfall variability within the monsoon season is also of importance to the predominantly agrarian societies of West Africa and South Asia. Over the Sahel, the majority of precipitation occurs from intermittent mesoscale convective complexes (MCS), embedded within westward propagating synoptic African Easterly Waves (Mathon et al., 2002), with a clear peak in precipitation variability on the 2-8 day time scale (Kiladis et al., 2006). Similarly over the SAM

region, a significant amount of rainfall is associated with synoptic-scale monsoon depressions that develop over the Bay of Bengal, before propagating northwestward across India and eventually dissipating over northwest India or Pakistan (Hunt et al., 2016). To assess the ability of GCMs to accurately simulate this synoptic rainfall we follow the approach described in the previous section and apply a 3-10 day band-pass filter to model and observed precipitation to highlight variability at the time scales of interest.

It is becoming increasingly established that higher model resolution provides a more realistic representation of the underpinning processes controlling weather and precipitation variability (e.g. Dawson and Palmer, 2015;Demory et al., 2014;Jung et al., 2012). In order to assess the benefit higher model resolution brings to the simulation of sub-seasonal precipitation variability over the WAM and SAM, in this section we analyze one of the EMBRACE models (EC-Earth) run in AMIP mode for the period 1980-2009, sampling atmospheric model horizontal resolutions of; T159 (128 km), T255 (80

20  km), T319 (64 km), T511 (40 km), T799 (25 km), T1279 (16 km), with a common set of 91 vertical levels. Findings from this analysis may offer pointers to an optimal resolution for other models to aim at with respect to simulating sub-seasonal precipitation variability as well as seasonal mean rainfall.

Figure 12 shows 3-10 day band-pass filtered precipitation variance for JJAS over Africa from two observational data sets (TRMM 3B42 and GPCP-1DD) and the six EC-Earth resolutions. The two observations differ markedly with respect to the

absolute magnitude of variance on these time scales. This is partly expected as the observational datasets feature a rather different horizontal resolution (0.25° vs. 1°). They do, however, exhibit some agreement in the spatial distribution of maxima and minima in precipitation variability, with a broad region of high variability stretching from Sudan west across to the Atlantic coast. GPCP-1DD indicates a northerly maximum in variability over West Africa around 12° N, associated with AEWs growing on the northern flank of the AEJ. Both data sets indicate a maximum in variability at the Atlantic coast,

around 10° N, and also relative maxima over the Ethiopian Highlands and Darfur mountains. In EC-Earth precipitation variability increases (and improves compared to the observations) as model resolution increases from T159 to T511. Beyond T511 there is little further increase in variability. In particular, as resolution increases from T159 to T511, higher variability appears eastwards back across the AEW wave track towards Ethiopia. There is also a clear increase in variability (wave





activity/intensity) at the Atlantic coast. Perhaps surprisingly, this increase in precipitation variability is not seen in the 850hPa meridional wind variability (not shown), which is typical measure of the AEW activity. Meridional wind variability is well simulated at T159 resolution and largely does not change right up to T1279. Hence, the increased model resolution seems to impact directly on moist processes that lead to rainfall on the ground, while having only minimal impact on the

dynamical structure of the AEWs. It is also worth noting that the seasonal mean (JJAS) precipitation changes very little across the EC-Earth resolutions (not shown), suggesting at lower resolutions (below T319), seasonal mean precipitation in EC-Earth, while relatively accurate, is made up of incorrect higher time frequency (weather) variability/intensities.

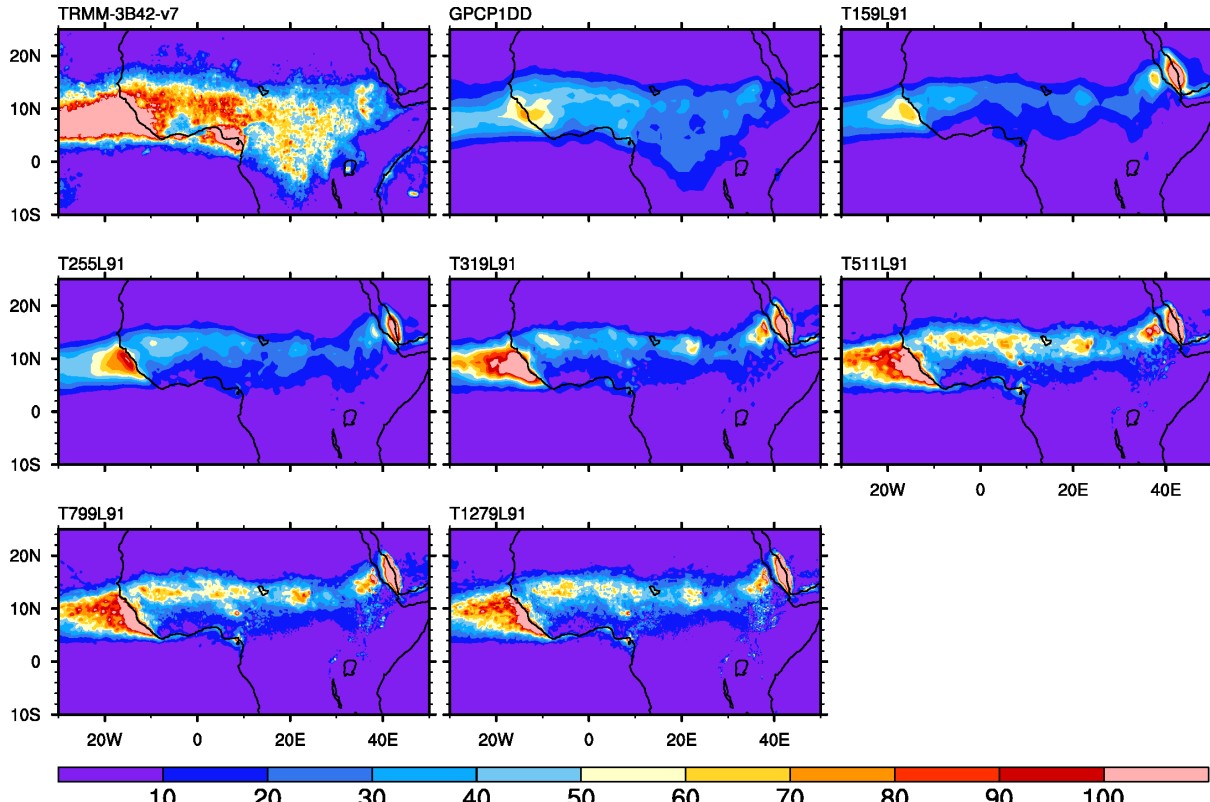

**Figure 12. 3-10 day band-pass filtered JJAS precipitation variance for TRMM, GPCP-1DD and T159, T255, T319, T511, T799,**
**T1279 EC-Earth simulations (mm$^2$ day$^{-2}$).**

Figure 13 shows 3-10 day filtered JJAS precipitation variance over the extended SAM region, again both TRMM and GPCP-1DD observations are plotted. As over the WAM region, variability is significantly higher in TRMM than GPCP with this being particularly the case over the equatorial Indian Ocean. Also similar to the WAM, precipitation variability increases (and improves) in EC-Earth as model resolution increases from T159 to T511, with little change thereafter. This increase is

true for variability associated with the SAM itself, but is not the case for variability over the equatorial Indian Ocean, which in fact slightly decreases (and degrades) as resolution increases beyond T255. As with the WAM, there is only a slight change (an increase) in seasonal mean (JJAS) precipitation in EC-Earth with increasing model resolution (not shown).





Although in regions of steep topography (such as the foothills of the Himalayas), there is an increase (improvement) in seasonal mean precipitation as model resolution increases.

**Figure 13. 3-10 day band-pass filtered JJAS precipitation variance for TRMM, GPCP-1DD and the T159, T255, T319, T511, T799, T1279 EC-Earth simulations (mm2 day-2).**

There is some suggestion of improvement in the representation of cloud-radiation interactions over the WAM region in moving from CMIP5 models to EMBRACE-updated models, with an impact on the large-scale dynamical structures over the region. Unfortunately, these bias reductions do not lead to clear improvements in regional rainfall (e.g. over the Sahel) or in rainfall variability. As with the SAM a major improvement in the representation of moist convection and its forcing of, and interaction with, clouds, radiation and the surface energy budget appears to be the most important requirement for a major



advance in the simulation quality of the WAM in present-day GCMs. Analysis of EC-Earth AMIP simulations at different model horizontal resolutions indicate an improvement in synoptic timescale precipitation variability as resolution is increased up to T511. This improvement occurs over both the WAM and SAM regions and, in both cases, seasonal mean rainfall is largely unchanged, suggesting mean WAM and SAM rainfall in lower resolution models (in the case of EC-Earth

lower than T511) may be correct but that this mean rainfall is composed of an incorrect underlying variability/intensity distribution.

## 3.3 Coupled tropical ocean climate

In the tropical Pacific the dominant easterly trade winds induce oceanic upwelling along the equator, resulting in a cold equatorial tongue of surface waters stretching from the coast of Central America to the date line. This cold tongue inhibits

deep atmospheric convection which becomes confined to west of ~170° E in the equatorial Pacific. In combination with the easterly trade winds and cold tongue, the mean equatorial ocean thermocline tilts from shallow depths in the East Pacific (mean 20 °C isotherm located at ~50 m depth) to deeper values (mean 20 °C isotherm at ~200 m) in the West Pacific. This coupled feedback, referred to as the Bjerknes feedback (Bjerknes, 1969;Neelin and Dijkstra, 1995) plays a key role in determining the mean state of the equatorial Pacific climate, as well as the main modes of variability around this mean state,

such as the El Niño Southern Oscillation (ENSO) (Bellenger et al., 2014). Similar coupled interactions, smaller in magnitude, also play a role in shaping the mean state of the tropical Atlantic (Xie and Carton, 2004).

Accurately representing the processes underpinning the mean state of the coupled tropical ocean is an important requirement of global climate models, necessary for confidence in their projections of future changes in both the mean state and ENSO variability, with changes in the latter being sensitive to small, systematic errors in the mean state (Bellenger et al.,

2014;Guilyardi, 2006). An accurate coupled mean state may also be important for simulating longer timescale variability in tropical ocean heat uptake (England et al., 2014;Meehl et al., 2011).

We implemented a number of performance metrics, developed by Li and Xie (2014), into the ESMValTool and used them to assess the ability of the EMBRACE AMIP and coupled models to simulate the coupled equatorial Pacific climate.

Figure 14a shows latitude cross-sections of DJF zonal mean precipitation from the AMIP simulations. Zonal means are for

all ocean grid cells between 120° E to 100° W. Observed SST is from HadISST (Rayner et al., 2003) and precipitation from CMAP from (Xie and Arkin, 1997), GPCP (Adler et al., 2003;Huffman and Bolvin, 2012) and TRMM (Huffman et al., 2007). For AMIP simulations all SST fields match the observations by design, except for HadGEM3-A which deviates slightly due to using daily SST and sea ice fields from Reynolds et al. (2007). Observed SSTs have a relative minimum on the equator, ~0.5 °C cooler than the SSTs at 7-8° S and ~1 °C cooler than SSTs at 7-8° N. Maximum SSTs are north of the

equator, ~0.5 °C warmer than at similar latitudes south of the equator. Observed precipitation shows a distinct maximum at ~8° N, with values of 7 mm day$^{-1}$ (GCPC) to 8 mm day$^{-1}$ (CMAP, TRMM). A second, weaker maximum (3-4 mm day$^{-1}$ depending on the observational data set) is seen at 8° S. A precipitation minimum is located on the equator, coincident with



the SST minimum. The AMIP models reproduce this structure of precipitation, with only small deviations from observations.

A different picture emerges for the coupled models (Figure 14b). All models, apart from HadGEM3-GC2, exhibit a widespread cold SST bias across the tropical Pacific, including a significant cold bias in the SST minimum at the equator.

This cold bias, however, has been substantially improved in the coupled EMBRACE simulations with EC-Earth3 and HadGEM3-GC2-N96. HadGEM3-GC2 has accurate SSTs, both north of and along the equator, but it exhibits a slight warm bias south of the equator and therefore fails to reproduce the north-south asymmetry in SST across the equator. This impacts the precipitation distribution in HadGEM3-GC2, with two maxima of similar magnitude symmetric about the equator and coincident with the model SST maxima. In contrast, EC-Earth3, while having a distinct cold bias along and south of the

equator, captures the south-north increase in SST across the equator. This meridional SST gradient appears crucial for capturing the observed asymmetry in precipitation, which EC-Earth3 successfully does. Both MPI models have a large SST cold bias in the tropics, particularly along the equator and simulate an ITCZ on either side of the equator. Comparing EC-Earth3 with HadGEM3-GC2, with respect to capturing the south to north increase in precipitation across the equator, it seems more important that models capture the corresponding gradient in SST than the absolute magnitude of equatorial

SSTs. Recent studies (e.g. Frierson et al., 2013;Marshall et al., 2014) suggest the overturning ocean circulation is responsible for a net transport of energy from the southern to the northern hemisphere, leading to the observed SST maximum being north of the equator. Kang et al. (2009) and Frierson and Hwang (2012) further argue that the location of the ITCZ, marking the low-level convergence of northern and southern hemisphere Hadley cells, is a direct result of this ocean-induced asymmetry in hemispheric energy and the southward directed, cross-equatorial upper branch of the Hadley cell balances the

northward ocean energy transport.





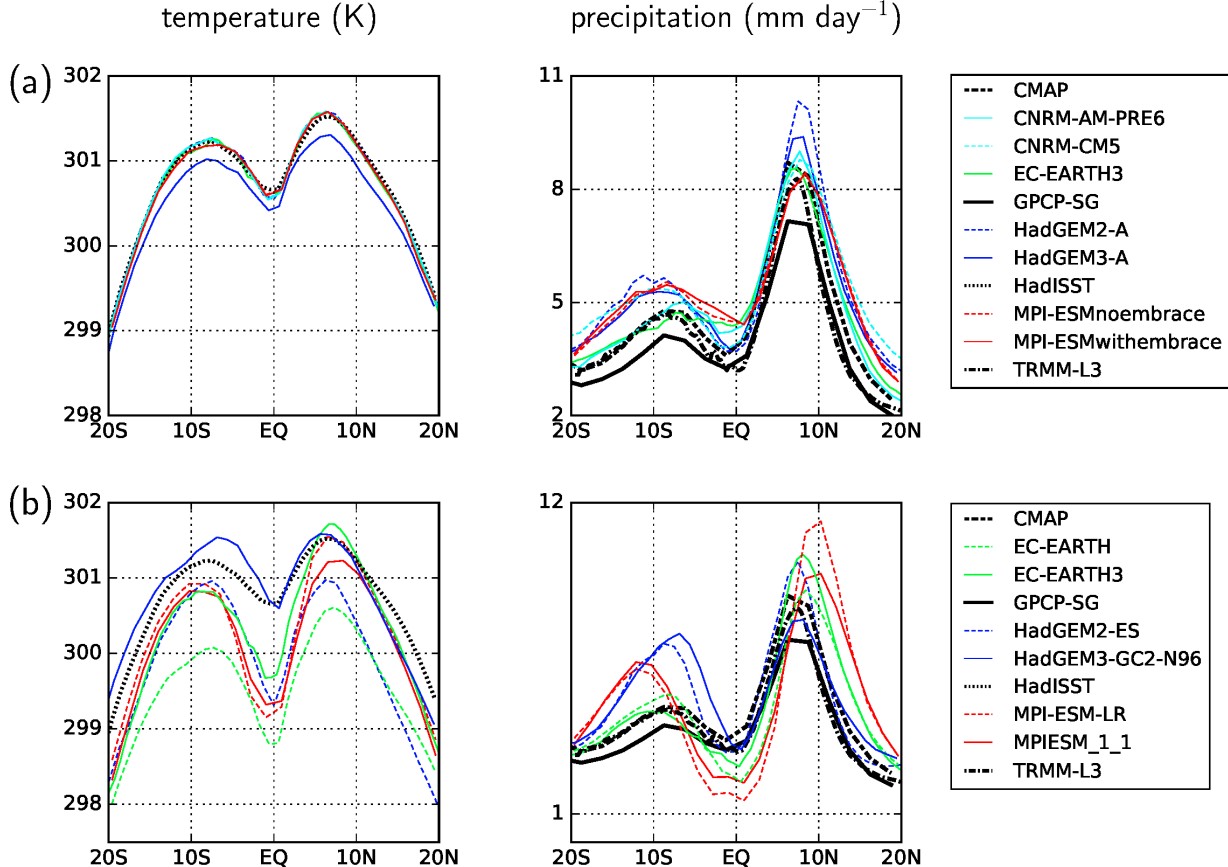

**Figure 14. Latitude cross-section of DJF SST in K (left column) and precipitation in mm per day (right column) from (a) the AMIP simulations and (b) the coupled (historical) simulations in comparison with observations (HadISST for SST and CMAP, GPCP and TRMM for precipitation). Values are zonal means averaged over all ocean grid cells in the longitude band 120° E to 100° W and averaged over the years 1982-2002 (MPI AMIP models 1979-1999).**

Similar findings hold for the tropical Atlantic (not shown). Observed SSTs are maximum at ~4° N, although there is a less distinct minimum along the equator. Precipitation is also maximum north of the equator. All coupled models, apart from HadGEM3-GC2, again show a systematic cold SST bias throughout the near-equatorial Atlantic. As in the Pacific, HadGEM3-GC2 has relatively accurate absolute SSTs, but a warm bias south of the equator so it does not simulate the south to north increase in SST. This leads to two ITCZ precipitation maxima symmetric about the equator in this model. EC-Earth3 again has a general cold SST bias, but correctly simulates the south to north gradient in SST and as a result also correctly simulates a single ITCZ rainfall maximum north of the equator.

In Figure 15 we follow the approach from Li and Xie (2014) to analyze the longitudinal structure of the coupled climate simulated in the equatorial Pacific. Figure 15 shows zonal mean precipitation, SST, 1000hPa zonal wind speed and 925hPa wind divergence, all averaged between 2.5° N and 2.5° S, from 120° E to 80° W. As not all ocean data were saved in the EMBRACE simulations, the depth of the ocean 20 °C isotherm (as used in Li and Xie (2014)) cannot be plotted and is replaced by 925hPa wind divergence.





Most AMIP models (left column in Figure 15) reproduce the zonal structure of precipitation across the Pacific, with minimal values from 80° W to 150° W and then an increase to a maximum in the West Pacific warm pool region ~145° E to 165° E. All models, with the possible exception of HadGEM3-A and CNRM-AM-PRE6 simulate too strong easterly trade winds (too negative values in Figure 15), particularly west of 160° W. In the AMIP models, this wind bias cannot impact the prescribed

SSTs. In the same cross-sections for the three coupled simulations (right column) only HadGEM3-GC2 has an accurate zonal structure of SST. All other models have a cold bias of 1 °C or more across the central Pacific (between 100° W to 170° E). Most models also underestimate precipitation in the equatorial band from 150° W to 160° E and feature a West Pacific rainfall maximum displaced 10-20° west of the observed maximum. Only HadGEM3-GC2, and to a lesser extent HadGEM2-ES, capture the correct zonal pattern of precipitation and location of the West Pacific maximum, indicating the

important role of SST for the zonal structure of precipitation. Both EC-Earth models show a significant easterly wind speed bias across most of the Pacific, as does HadGEM2-ES east of 150° W. Excess 925hPa wind divergence is seen in all three coupled models. HadGEM3-GC2 has the most accurate simulation of equatorial zonal wind speeds and wind divergence. This suggests that the two EC-Earth models and HadGEM2-ES, excessive easterly winds induce too strong Ekman divergence and ocean upwelling along the equatorial Pacific leading to the cold SST bias. The two MPI-ESM models have

significant cold SST biases across the Pacific (2 °C) even though the simulated zonal wind speeds are relatively accurate, contrasting significantly with the two MPI-ESM AMIP models, in which the largest (positive) easterly wind biases are seen. The cold SST bias in MPI-ESM is accompanied by a significant positive bias in 925hPa wind divergence (excessive low-level equatorial divergence), indicating too strong meridional (poleward directed) wind components near the equator in this model. The findings suggest excess surface momentum loss from the easterly trade winds, driving both a cold SST bias

along the equator and excessive poleward directed winds just off the equator in the MPI coupled models.




**AMIP**                                                    **coupled**



**Figure 15. Equatorial mean (2.5° N to 2.5° S) values of from top to bottom: precipitation, SST, 1000hPa zonal wind speed and total 925hPa wind divergence. Values are plotted for the equatorial Pacific between 120° E and 80° W. The left column shows model results from the AMIP simulations, the right column from the coupled (historical) simulations in comparison with observations.**

5   Simulated moist convection over the tropical oceans is extremely sensitive to small errors (~0.5 °C) in both the absolute value, and the spatial gradient, of SSTs near the equator. HadGEM3-GC2 has the most accurate absolute value of tropical SSTs, but suffers from a double-ITCZ problem due its meridional SST gradients across the equator being incorrect. In contrast, EC-Earth3 has a systematic cold SST bias in both the equatorial Pacific and Atlantic but captures the correct meridional gradient in SST between the two hemispheres. As a result EC-Earth3 does not exhibit a double-ITCZ, with clear

10   precipitation maximum north of the equator in both basins collocated with SST maxima. Two of the three EMBRACE





models (HadGEM3-GC2 and EC-Earth3) show improvement in simulated tropical SSTs compared to their CMIP5 versions. HadGEM3-GC2 in particular, has a very accurate zonal structure of SST across the equatorial Pacific, along with associated atmospheric phenomena (precipitation, easterly trade winds). EC-Earth3 also shows some improvement over its CMIP5 version.

**3.4 Southern Ocean clouds and radiation**

The Southern Ocean plays a key role in the earth's climate, being one of the few extensive regions of the globe where the deep ocean is in regular contact with the surface, allowing significant atmosphere-ocean exchange of heat (Kuhlbrodt and Gregory, 2012) and $CO_2$ (Frölicher et al., 2015). Further south, formation of Antarctic deep water efficiently transports surface waters into the deep ocean. Both these phenomena are key components of the global ocean overturning circulation
(Marshall and Speer, 2012). Trenberth and Fasullo (2010) show that current GCMs have a persistent underestimate in reflected shortwave (SW) radiation at the top of the atmosphere (TOA) over the Southern Ocean, implying too much SW radiation reaching the ocean surface. Linked to this many coupled GCMs also show a warm SST bias over extensive parts of the Southern Ocean. This bias increases the vertical stability of the upper ocean and can therefore impact the overturning ocean circulation. Trenberth and Fasullo (2010) and Sallée et al. (2013) suggest such biases compromise the reliability of
climate change projections in the region.

To assess GCM simulated surface energy budgets over the Southern Ocean, a number of metrics have been implemented into the ESMValTool. In this section we apply some of these metrics to assess the EMBRACE models' ability to capture phenomena controlling the surface radiation budget of the Southern Ocean. We focus on austral summer, when incoming surface radiation is at a maximum and model errors are generally the largest. We analyze total cloud amount, cloud liquid
(LWP) and ice (IWP) water path, and surface and TOA solar radiation. Analysis is only performed for the AMIP simulations as the main findings also apply to the coupled models.

Figure 16 shows cross-sections from 65° S to 30° S of simulated zonal mean DJF total cloud cover, LWP and IWP, TOA outgoing (SWUP) and surface downwelling (SWD) shortwave radiation compared to satellite observations. For LWP and IWP, ERA-Interim reanalysis data are also included as a second observationally based estimate (Dee et al., 2011). Observed
cloud cover increases from ~60% at 30° S to more than 90% around 60° S. Most models capture this poleward increase, although all, except HadGEM3-A, exhibit a systematic negative bias (of 5-15%) across the band ~45° S to 65° S. HadGEM3-A has the most accurate cloud cover and is a clear improvement over HadGEM2-A. CNRM-AM-PRE6 also shows improvement compared against its CMIP5 version. EC-Earth3 shows a small improvement, while the MPI-ESM model shows little change.
The impact of clouds on solar radiation can be summarized by cloud optical depth, which is a function of the cloud water content and the effective radius of cloud liquid droplets/ice crystals, integrated over cloud depth (Slingo, 1989). Here, vertically integrated LWP values are compared to observed estimates (LWP and IWP are not available from HadGEM3-A or EC-Earth). LWP observations are based on the University of Wisconsin (UWisc, O'Dell et al. (2008)) satellite passive



microwave data set, IWP observations are MODIS collection 6 data (Platnick et al., 2003). Similar to Jiang et al. (2012), the MODIS IWP data representing in-cloud values have been multiplied with the observed ice cloud fraction for comparison with the grid-box averages provided by the models. Due to the large differences across remotely sensed LWP/IWP data sets, values from UWisc and MODIS should be viewed as indicative at best. We include LWP/IWP estimates from ERA-Interim as a second constraint to provide some measure of this uncertainty. Our main motivation is to show the large range in both LWP and IWP across models, which may partly be due to the weak observational constraint.

South of ~40° S, LWP differs by a factor of ~2 across models, with IWP showing an even larger inter-model spread (up to a factor of ~3). Such large differences will clearly impact solar radiation fluxes. Before robust guidance on model biases in cloud water biases can be provided for the Southern Ocean, further work is needed to quantify the uncertainty/accuracy of LWP/IWP observations. For now we stress (i) the wide range of LWP and IWP across models and (ii) the lack of a robust observational constraint on these two variables.

Observed SWUP also increases southwards, paralleling the increase in observed cloud cover (Figure 16a,c). The spread in both SWUP and SWD is decreased going from CMIP5 to the updated models. This is likely primarily a result of the reduced spread (and reduced bias) in cloud cover in the updated models. Nevertheless, a negative bias in SWUP (too little SW reflection) of ~10-40 Wm$^{-2}$ is still seen for all four updated EMBRACE models south of ~55° S. This translates into a positive bias in SWD of similar magnitude over the same region. The underestimate in SW reflection for most models is consistent with the (~5-10%) underestimate of cloud cover south of 55° S (only HadGEM3-A does not have a negative bias in cloud cover in this region). The SWUP negative bias is also consistent with an implied underestimate of LWP in EC-Earth3 and CNRM-AM-PRE6 if UWisc data are used as the observational constraint.



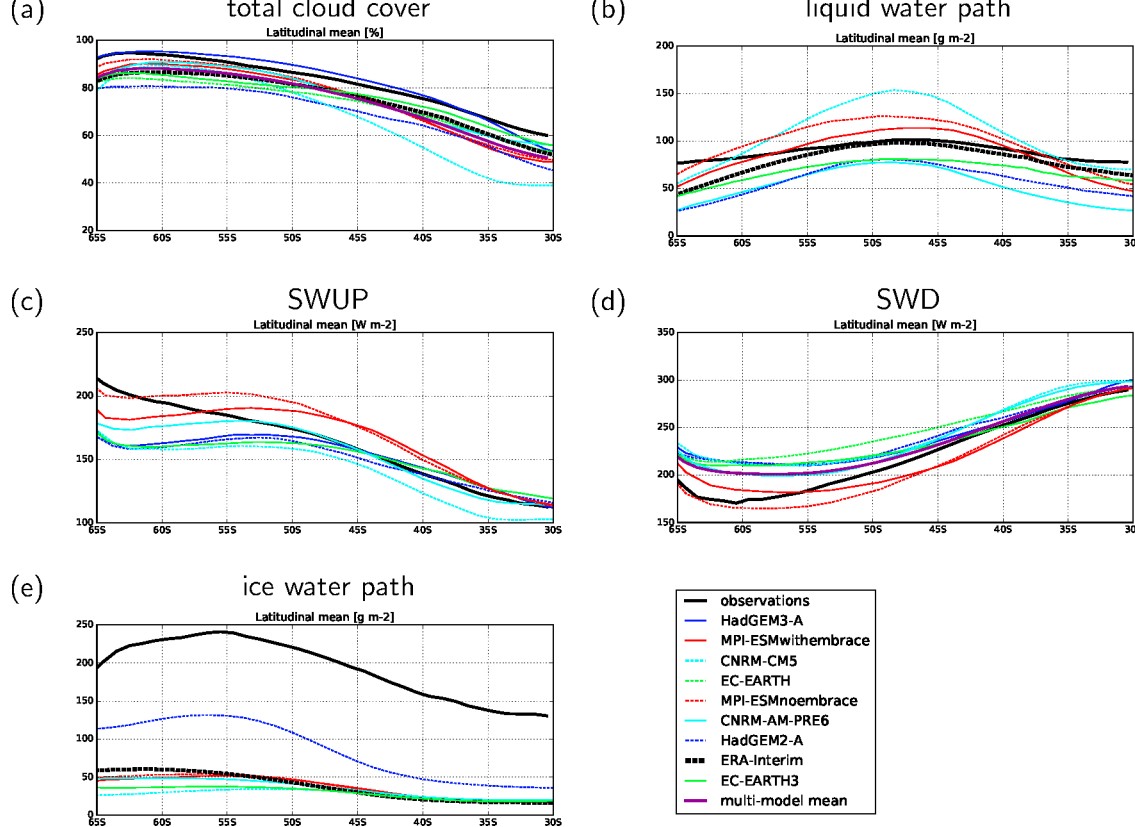

**Figure 16. Latitude cross-section of DJF zonal mean (a) total cloud cover, (b) liquid water path, (c) TOA outgoing shortwave radiation (SWUP), (d) surface downwelling shortwave radiation (SWD), and (e) ice water path. Shown are the CMIP5 and EMBRACE coupled simulations in comparison with satellite observations and the ERA-Interim reanalysis: (a) and (e) MODIS, (b) UWisc, (c) and (d) CERES-EBAF.**

To gain more insight into the relationship between cloud cover and reflected SW, Figure 17 shows scatter plots of monthly mean TOA SWUP and surface SWD each plotted against monthly mean cloud cover for all available DJF months over 20-year simulation period. Observations are from CERES-EBAF (2001-2014) SWUP/SWD and MODIS-L3 cloud cover (2003-2014). The figure is constructed as follows: for each ocean grid point in the band 30° S to 65° S, monthly cloud cover is binned into 5% width bins (from 0 and 100%) and for each cloud cover occurrence the corresponding SWUP/SWD is saved to that bin. This is carried out for all grid points and DJF months, resulting in a mean DJF SWUP/SWD value for each of the 20 cloud cover bins and scatter plots of SWUP/SWD as a function of cloud cover for the region 30° S to 65° S. The fractional occurrence of cloud cover amounts for each 5% bin were also recorded and plotted as a frequency distribution (Figure 17, middle row).

The observed cloud cover histogram shows the bulk of months have cloud cover >80%. Most models capture this distribution, with clear improvements in the updated versions of the CNRM and HadGEM models. EC-Earth3 underestimates the occurrence of cloud cover >90%. For the SWUP-cloud cover scatter plots, most models underestimate



SWUP (and linked to this overestimate SWDN) for cloud cover < 50%, although the fractional occurrence of cloud cover <50% is extremely low (middle row in Figure 17), so this bias may arise from poor sampling and will have minimal impact on the zonal mean SWD and SWUP biases in Figure 16. All models overestimate SWUP for cloud cover >60% (the most frequently occurring cloud amount). These biases range from ~25-30 Wm$^{-2}$ (MPI-ESM) to ~5 Wm$^{-2}$ (HadGEM3-A) and are

generally coincident with an underestimate of SWD for the same cloud cover amounts. This finding is not consistent with the zonal mean SWUP and SWD biases seen in Figure 16, particularly south of ~50° S, where all the models underestimate TOA SWUP and overestimate surface SWD, ranging from ±10-30 Wm$^{-2}$.

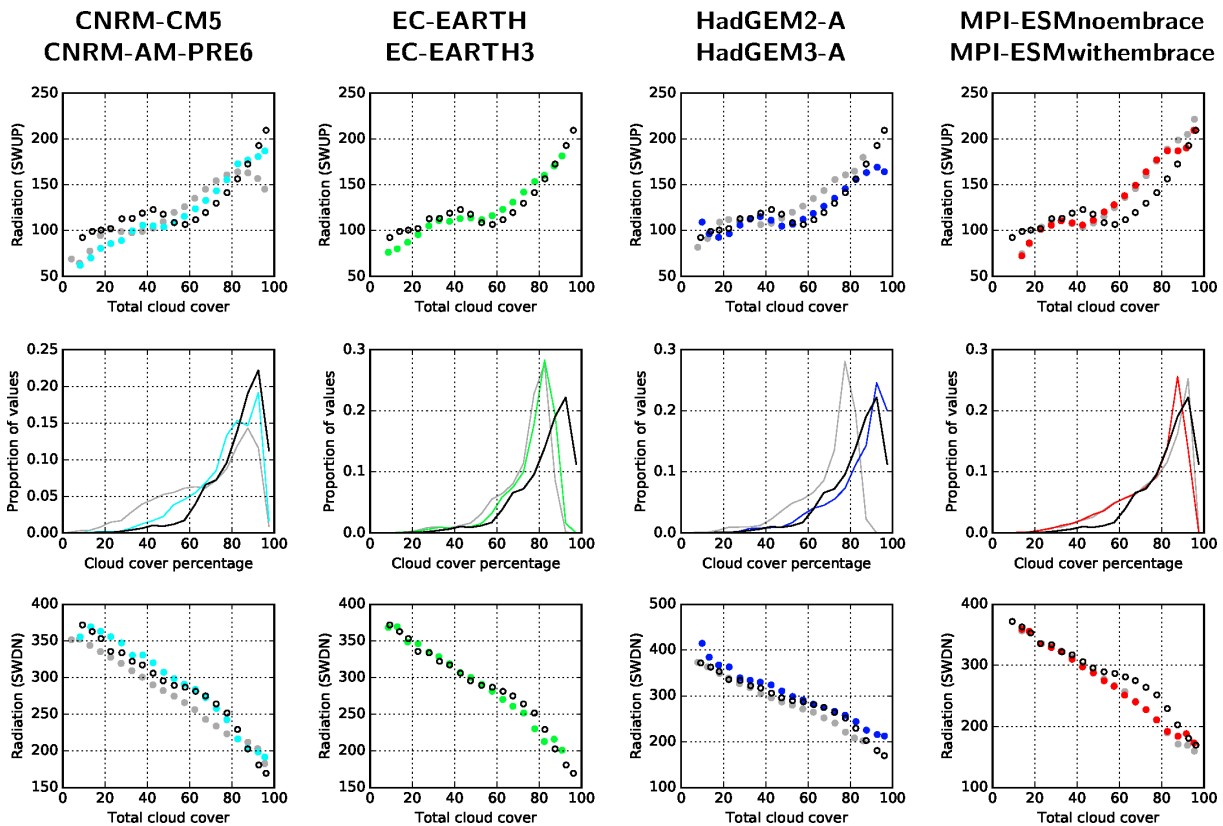

**Figure 17. Top row: scatterplot of monthly mean TOA SWUP versus total cloud cover for the Southern Ocean region 30° S-65° S**
**and season DJF. Bottom row is the same for surface SWDN. Middle row shows fractional occurrence of monthly mean cloud cover over this region. Cloud observations are MODIS-L3 and SWUP/SWDN is from CERES-EBAF. The EMBRACE-updated AMIP models are plotted in color, the corresponding CMIP5 models are shown in gray, observations are in black.**

To understand this inconsistency, Figure 18 and Figure 19 repeat the radiation-cloud histograms separately for the latitude bands 30-45° S (referred to below as SOC-N) and 50-65° S (referred to below as SOC-S). For SOC-N the observed cloud
histogram is shifted towards lower values, with a peak at 80% and a tail of occurrences down to 20%. HadGEM3-A captures this distribution as, to a lesser extent, does EC-Earth3. The other models all show too frequent cloud cover <60% and too little cloud occurrence >80%. The tendency for all models to have a positive bias in TOA SWUP for cloud amounts >50% is also seen in this region, although HadGEM3-A is quite accurate in this regard. Figure 16 indicates the updated EMBRACE





models have relatively small zonal mean TOA SWUP and surface SWDN errors in the band 30° S to 45° S. For the MPI-ESM and CNRM models this partly results from error cancellation, with an underestimate of cloud amount balanced by the most frequent cloud amounts (>50%) being too reflective. HadGEM3-A has an accurate simulation of zonal mean SWUP and SWD in this latitude band from both accurate cloud amounts and accurate SWUP/SWD-cloud cover relationships.

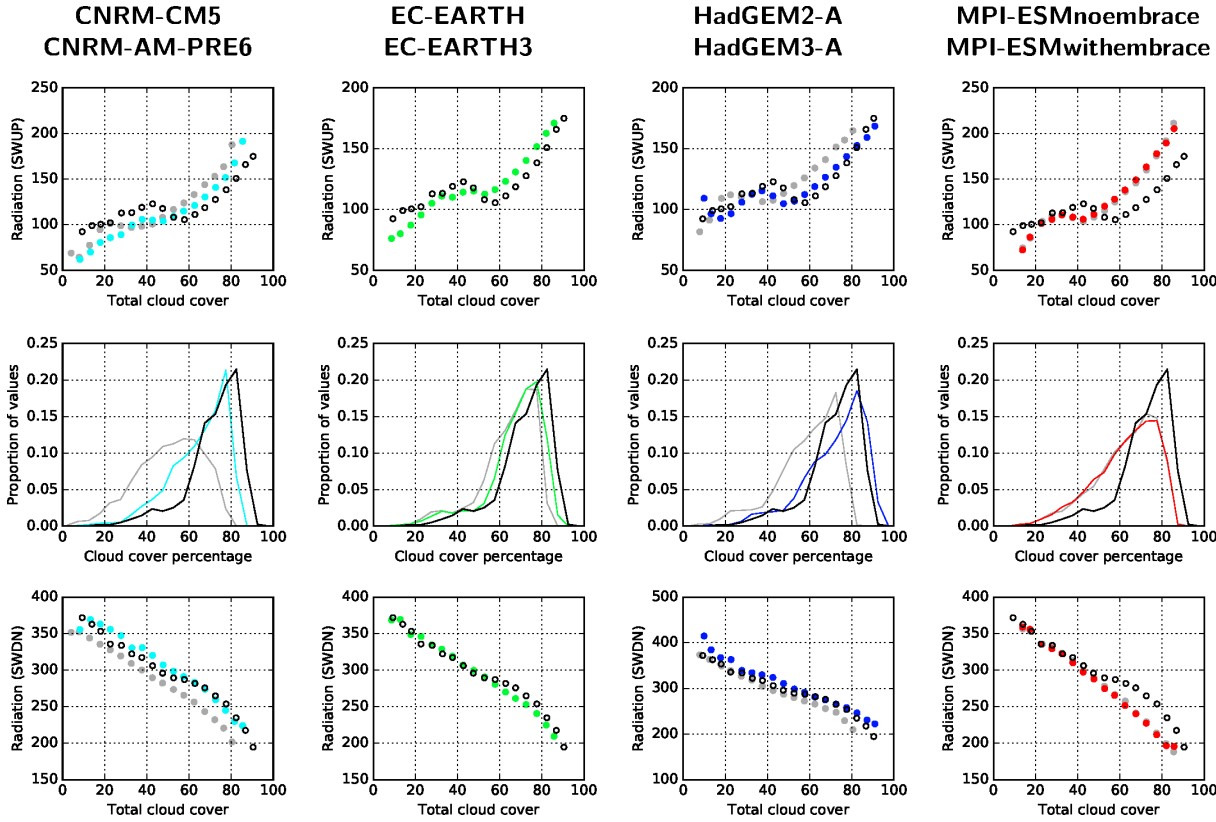

**Figure 18. As Figure 17 but for the northern part of the Southern Ocean region (SOC-N, 30° S-45° S).**

Over the SOC-S region, Figure 16 shows all updated models have a negative bias in zonal mean TOA SWUP and a positive bias in surface SWD. The SWUP/SWD-cloud cover scatter plots for this region show more mixed results (Figure 19). This may partly be due to a small sample size, although the main findings we believe are robust. The observed cloud histogram indicates monthly cloud cover >90% dominates at these latitudes. EC-Earth3 and, to a lesser extent, MPI-ESM underestimate the frequency of occurrence of such high cloud amounts and for these two models this is the leading cause of the negative/positive bias in the SWUP/SWD zonal means. While there is scatter in the observed SWUP/SWD-cloud cover relationship over SOC-S, EC-Earth3 and MPI-ESM capture the relationship quite well, suggesting clouds, when present in these two models in this latitude band, have an accurate representation of SW reflection and transmission. In contrast, CNRM-AM-PRE6 and HadGEM3-A do well simulating the cloud distribution, but have more mixed success capturing the observed SWUP/SWD-cloud relationship. CNRM-AM-PRE6 reproduces this relationship best of these two models.





HadGEM3-A reproduces the observed cloud cover histogram very well, but fails to reproduce the SWUP/SWD-cloud relationship, with an underestimate in TOA SWUP for cloud >95% of ~30-40 Wm$^{-2}$ and a similar error of opposite sign in SWD. This is the leading cause of the zonal mean SWUP/SWD biases in HadGEM3-A.

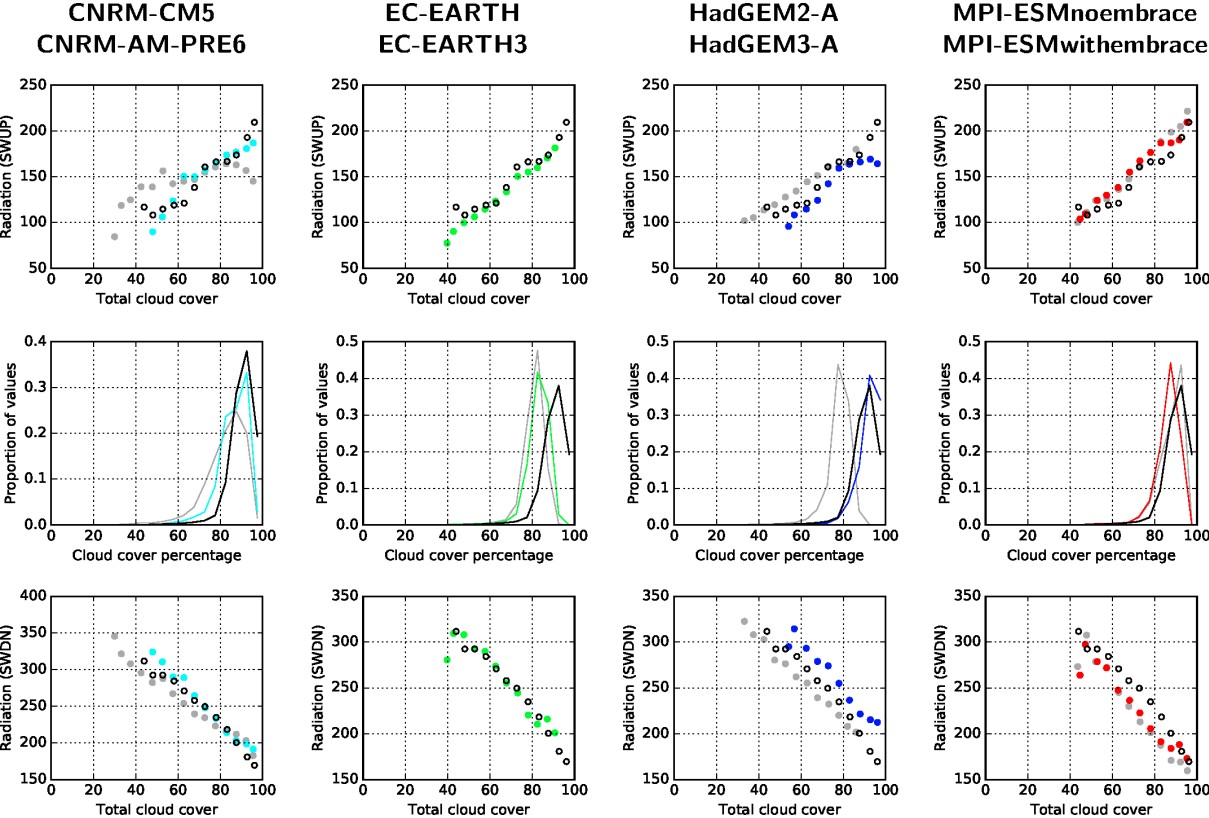

Figure 19. As Figure 17 but for the southern part of the Southern Ocean region (SOC-S, 50° S-65° S).

There is a clear improvement in cloud amounts simulated over the Southern Ocean in the majority of EMBRACE-updated models. This is particularly true for HadGEM3-A compared to HadGEM2-A where a systematic 10% underestimate of cloud cover is reduced to close to zero. The CNRM model also shows an improvement in cloud amounts across the Southern Ocean. SWUP and SWDN are also surprisingly well captured in most of the updated models, with only the MPI-ESM showing a systematic bias in SWUP (too much reflected SW radiation at TOA) and SWD (too little SWD at the surface) for cloud amounts >60%.

Three models show a tendency for compensating biases (too few clouds balanced by clouds being too reflective) resulting in accurate SWUP and SWD over the 30° S to 45° S band. Only HadGEM3-A captures both the cloud occurrence distribution SWUP/SWD-cloud relationship over this region. Further south (50° S to 65° S) most models (apart from EC-Earth3) capture the shift in most frequent cloud occurrence to >90%. In this region models have a greater problem simulating the SWUP/SWD-cloud relationship. For example, both HadGEM3-A and CNRM-AM-PRE6 have significant positive biases in



surface SWD for cloud amounts of 95-100%, likely related to an underestimate of cloud optical depth for these cloud types. For further improvement of cloud and radiation processes over the Southern Ocean improved observational constraints, particularly with respect to in-cloud constituents (e.g. liquid and ice water amounts) are required.

## 4 Discussion and conclusions

The tropical precipitation in three out of four EMBRACE models analyzed is significantly improved with wet biases in these regions reduced by up to 1-2 mm day$^{-1}$ compared with the CMIP5 simulations. Precipitation, in particular in tropical regions, remains challenging to model with large biases in the West Pacific and Indian Ocean as well as in the ITCZ and SPCZ. Two of the EMBRACE-updated coupled models exhibit significant improvements in tropical SSTs, while only one model (EC-Earth3) does not show a double-ITCZ in the Pacific.

Biases in the near-surface temperature climatology are still present over many parts of the tropical continents. For example, in most of the analyzed models, a warm bias over Central Africa and northern South America is found. In the coupled simulations, large biases are also still present in the Southern Ocean along the coast of Antarctica. This bias is consistent with the solar radiation biases seen in the four EMBRACE models south of 50° S.

    The ESMs still have significant problems in accurately simulating all features of the two large-scale atmospheric circulation

patterns, the South Asian and West African monsoons. Many of the problems can be traced to difficulties in accurately simulating moist convection over land and interactions between moist convection and (i) convectively-forced clouds and impacts on solar radiation and subsequently surface evaporation and soil water initially small biases in moist convection (e.g. in geographical location, intensity or temporal offsets within the diurnal cycle) can be amplified through these interactions, leading to systematic biases in seasonal mean values. (ii) Convective rainfall and its impact on surface-soil water amounts

and surface evaporation initially small biases can be amplified through feedback processes. For example, rainfall occurring too early in the diurnal cycle (a common bias in many GCMs) will allow a larger fraction of rainfall to be locally evaporated back into the atmosphere instead of percolating into the deeper soil and increasing total soil moisture amounts. A gradual drying out of the surface soil layer will induce upward percolation of soil water and a deepening of the drying signal. The result will be a drying out of soils and a reduced ability to locally sustain moist convection and rainfall, again leading to an

amplification of the original bias.

    Both feedback loops can be seen as local or regional processes. Once established, these biases can influence the large-scale (surface and mid-tropospheric) thermal gradients driving the monsoon circulation, pushing the simulated monsoon even further from the one observed. The representation of moist convection and its interaction with solar radiation (through convectively-forced clouds) and the land surface (through solar radiation and precipitation) are therefore key

parameterizations requiring improvement for significant progress in simulating both the South Asian and West African monsoons.





Some improvements are seen in both the South Asian and West African monsoon from the EMBRACE models compared with their CMIP5 versions. However, significant biases remain, particularly with respect to regional rainfall patterns and the annual cycle of monsoon rainfall. Even more significant biases are seen for intra-seasonal rainfall variability, with little progress from CMIP5 models. In the three coupled model SAM simulations, biases in precipitation and monsoon circulation

(given by the 850hPa wind field) are reduced compared to their CMIP5 counterparts. The primary reason for this is coupled feedbacks that enable the damping of an atmospheric error (e.g. in wind speed or atmospheric moisture content) through the introduction of a compensating bias in surface ocean temperatures (e.g. a cold SST bias). The main model bias regarding West Africa relates to higher time frequency precipitation variability on time scales associated with African Easterly Waves. These systems, and the convective complexes embedded within them, deliver the majority of rainfall to the West African

Sahel. A realistic simulation of AEWs seems an important prerequisite for increasing confidence in future rainfall projections over the Sahel. Most of the EMBRACE models and their CMIP5 versions have severe difficulty in simulating these waves, with little improvement from CMIP5 to the EMBRACE-updated models. The models show quite some spread in their ability to simulate near-surface temperatures over the Sahara, with JJAS mean differences of up to 5 ℃ across models. Given the importance of the Saharan heat low in the overall West African monsoon circulation, more emphasis on

simulating the surface energy budget over the Sahara seems necessary. All coupled simulations over West African suffer from excess precipitation at the Guinea coast. This is a direct result of a warm SST bias in all models off the coast of Namibia-Angola. Reduction of this systematic bias, likely through improved ocean physics, resolution, and an improved simulation of marine stratocumulus clouds, will be a necessary step for improving coupled simulations of the West Africa monsoon.

Analysis of AMIP-type simulations performed with EC-Earth at different horizontal resolutions of up to T1279 show an improvement (i.e. increase) in the variability of precipitation on the synoptic time scale with increasing horizontal resolution up to T511. The seasonal mean rainfall over the WAM and SAM regions, however, does not change significantly with horizontal resolution suggesting that the reasonably good agreement of modeled and observed mean WAM and SAM rainfall in lower resolution models may be based on an unrealistic variability/intensity distribution.

Many models suffer from an excessive cold tongue of water along the Pacific equator, with this tongue being both too cold and extending too far into the West Pacific. Combined with this cold tongue, coupled models also typically show (i) too strong easterly trade winds along the equator, (ii) equatorial rainfall shifted too far west in the West Pacific, (iii) an equatorial thermocline that is too shallow in the East Pacific and too deep in the West Pacific, and (iv) a double-ITCZ, often with excess rainfall south of the equator. Comparison of the three EMBRACE coupled models show a general tendency for

improved equatorial SSTs, both in the Pacific and Atlantic. HadGEM3-GC2 and EC-Earth3 show significant improvement in SST bias (of as much as 1 ℃ in the zonal and DJF seasonal mean). HadGEM3-GC2, in particular, has a very accurate simulation of tropical SSTs and does not appear to suffer from an excessive equatorial Pacific cold tongue. This is a clear improvement over HadGEM2-ES and is an important reduction in a systematic bias. In combination with the SST improvement HadGEM3-GC2 also shows a clear improvement in the strength of the easterly trade winds along the equator.



This is likely the primary cause of the reduced SST bias (through reduced Ekman driven upwelling along the equator). SSTs in EC-Earth3 are also improved relative to EC-Earth used in CMIP5. Although not as accurate in an absolute sense as HadGEM3-GC2, the meridional structure of SST around the equator is better in EC-Earth3. This improved spatial structure plays an important role in EC-Earth3 not exhibiting a double-ITCZ, with an accurate northern hemisphere maximum in

precipitation in both the Pacific and Atlantic. Along the equatorial Pacific, EC-Earth3 still suffers from a systematic cold bias (although improved relative to the CMIP5 version of EC-Earth), accompanied by too strong easterly trade winds.

Most of the EMBRACE-updated models show a clear improvement in monthly cloud cover over the Southern Ocean compared to their CMIP5 predecessors. These improvements feed through into reduced bias (and inter model spread) of both TOA outgoing solar radiation and surface downwelling solar radiation. A reduction in inter-model spread is also seen for

liquid water path suggesting that the reduced spread translates into reduced model bias, although the observations of LWP over the Southern Ocean suffer from high uncertainties. All four EMBRACE-updated AMIP models have a negative bias in SWUP south of 50° S increasing to -20 to -40 $Wm^{-2}$ in the 60° S to 65° S band. A similar magnitude positive bias in SWD is seen in the same region. While the models show quite some improvement over their CMIP5 counterparts, the SWUP/SWD biases will drive a warm SST bias in the Southern Ocean, south of the Antarctic Circumpolar Current (ACC) with negative

effects on vertical upwelling and, further south Antarctic deep water formation and sea-ice amounts.

The main outstanding cloud-radiation biases appear to be in the southernmost region of the Southern Ocean (e.g. increasing with increasing southerly latitude from 50° S). Whether this highlights problems that are specific to certain cloud types (e.g. mid-level clouds in the cold sector of mid-latitude weather systems (Bodas-Salcedo et al., 2012)), problems in correctly delineating between liquid, solid and supercooled cloud water (Lawson and Gettelman, 2014) or problems simulating cloud

formation in a relative pristine (natural aerosol dominated) region (McCoy et al., 2015) requires further analysis and, in particular, more robust observational constraints.

## 5 Code and data availability

This analysis has been done with the ESMValTool, which is released under the Apache License, VERSION 2.0. The newly added ESMValTool namelist 'namelist_lauer17james.xml' includes the diagnostics that can be used to reproduce the figures

of this paper. This version will be available from the ESMValTool webpage at http://www.esmvaltool.org/ and from github (https://github.com/ESMValTool-Core/ESMValTool). Users who apply the software resulting in presentations or papers are kindly asked to cite the ESMValTool documentation paper (Eyring et al., 2016b) alongside with the software doi (doi: 10.17874/ac8548f0315) and version number. The climate community is encouraged to contribute to this effort and to join the ESMValTool development team for contribution of additional diagnostics for ESM evaluation. Data from the CMIP5 models

is publically available through the Earth System Grid Federation (ESGF), the EMBRACE model runs can be made available on request from the host modeling groups.





## Acknowledgements

This work has been performed and funded within the European Commission's 7th Framework Programme, under Grant Agreement number 282672, the "Earth system Model Bias Reduction and assessing Abrupt Climate change (EMBRACE)" project, and continued within the European Union's Horizon 2020 research and innovation programme under grant

agreement No. 641816 (CRESCENDO), and the DLR project "Klimarelevanz von atmosphärischen Spurengasen, Aerosolen und Wolken: Auf dem Weg zu EarthCARE und MERLIN (KliSAW)". We acknowledge the World Climate Research Program's (WCRP's) Working Group on Coupled Modelling (WGCM), which is responsible for CMIP, and we thank the climate modelling groups for producing and making available their model output. For CMIP5 the US Department of Energy's Program for Climate Model Diagnosis and Intercomparison provides coordinating support and led development of

software infrastructure in partnership with the Global Organization for Earth System Science Portals. R. Roehrig acknowledges the work of the CNRM climate model development team, which contributed to the development of the CNRM-AM-PRE6 prototype version used in the present study. He also acknowledges the support by the French national program LEFE/INSU under the DEPHY2 project.

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



**Table 1. List of models analyzed. The improved models can be close to the upcoming CMIP6 versions or intermediate versions, which can include both, EMBRACE specific and general improvements.**

| Model | Atmosphere | Land and Vegetation | Ocean and Sea Ice | Reference |
|---|---|---|---|---|
| **CMIP5:** CNRM-CM5 | Arpege v5, T127L31 (~1.4°) | Surfex v5 | --- | Voldoire et al. (2013) |
| **Improved:** CNRM-AM-PRE6 | Arpege v6, T127L91 (~1.4°) | Surfex v7.3 | --- | Michou et al. (2015) |
| CMIP5: MPI-ESM | ECHAM6.1, T63L47 (~1.875°) | JSBACH 2.0 | MPIOM, GR15L40 (~1.5°) | Stevens et al. (2013), Jungclaus et al. (2013) |
| **Improved:** MPI-ESM | ECHAM6.3, T63L47 (~1.875°) bug fixes and re-calibration of cloud processes | JSBACH 3.0 5-layer hydrology, YASSO soil carbon model | --- | Hagemann and Stacke (2015), Goll et al. (2015) |
| **CMIP5:** EC-Earth | IFS Cycle 31, T159L62 (~1.125°) | HTESSEL | NEMO v2, ORCA1L42 | Hazeleger et al. (2013) |
| **Improved:** EC-Earth3 | IFS Cycle 36r4, T255L91 (~0.70°) | HTESSEL | NEMO v3.3.1, ORCA1L46 | |
| **CMIP5:** HadGEM2-ES | HadGEM2-A, N96L38 (~1.25° x 1.875°) | MOSES2/TRIFFID | HadGEM2-O, ~1°, L40 | Martin et al. (2011), Collins et al. (2011) |
| **Improved:** HadGEM3-GC2 | GA6.0, N96L85 (~1.25° x 1.875°) | JULES GL6.0 | GO5.0 (NEMO v3.4), GSI6.0, ORCA0.25L75 | Williams et al. (2015), Walters et al. (2017), Megann et al. (2014), Rae et al. (2015) |



**Table 2. List of model configurations and model experiments analyzed. If more than one ensemble member is available, only the first ensemble "r1i1p1" is analyzed.**

| Model name | Generation | Atmosphere, horizontal resolution | Available model years |
|---|---|---|---|
| **Atmosphere-only (AMIP) experiments** | | | |
| **CNRM-CM5** | CMIP5 | 256x128 (~1.4° x 1.4°) | 1979-2008 |
| **EC-Earth** | CMIP5 | 320x160 (~1.1° x 1.1°) | 1979-2008 |
| **HadGEM2-A** | CMIP5 | 192x145 (~1.9° x 2.5°) | 1979-2008 |
| **MPI-ESMnoembrace** | CMIP5 | 192x96 (~1.9° x 1.9°) | 1979-1999 |
| **CNRM-AM-PRE6** | EMBRACE | 256x128 (~1.4° x 1.4°) | 1979-2012 |
| **EC-Earth3** | EMBRACE | 512x256 (~0.7° x 0.7°) | 1980-2014 |
| **HadGEM3-A** | EMBRACE | 192x144 (~1.9° x 2.5°) | 1982-2012 |
| **MPI-ESMwithembrace** | EMBRACE | 192x96 (~1.9° x 1.9°) | 1979-1999 |
| **Coupled (historical) experiments** | | | |
| **EC-Earth** | CMIP5 | 320x160 (~1.1° x 1.1°) | 1850-2009 |
| **HadGEM2-ES** | CMIP5 | 192x145 (~1.9° x 2.5°) | 1859-2005 |
| **MPI-ESM-LR** | CMIP5 | 192x96 (~1.9° x 1.9°) | 1850-2005 |
| **EC-Earth3** | EMBRACE | 512x256 (~0.7° x 0.7°) | 1980-2014 |
| **HadGEM3-GC2-N96** | EMBRACE | 192x144 (~1.9° x 2.5°) | 1950-2005 |
| **MPIESM_1_1** | EMBRACE | 192x96 (~1.9° x 1.9°) | 1980-2005 |
| **High-resolution (AMIP) EC-Earth experiments** | | | |
| **EC-Earth3-T159** | EMBRACE | 320x160 (~1.1° x 1.1°) | 1980-2009 |
| **EC-Earth3-T255** | EMBRACE | 512x256 (~0.7° x 0.7°) | 1980-2009 |
| **EC-Earth3-T319** | EMBRACE | 640x320 (~0.6° x 0.6°) | 1980-2009 |
| **EC-Earth3-T511** | EMBRACE | 1024x512 (~0.4° x 0.4°) | 1980-2009 |
| **EC-Earth3-T799** | EMBRACE | 1600x800 (~0.2° x 0.2°) | 1980-2009 |
| **EC-Earth3-T1279** | EMBRACE | 2560x1280 (~0.1° x 0.1°) | 1980-2009 |





**Table 3. Observationally based data sets used for the model evaluation. For variable definitions, see Table 4.**

| Data set | Type | Variable(s) | Resolution | Years | Estimate of systematic errors | Reference |
|---|---|---|---|---|---|---|
| CERES-EBAF | satellite | lw_cre, rsut, rsds, rlds, sw_cre | 1° x 1° | 2001-2012 | ~5 Wm$^{-2}$ | (Loeb et al., 2012;Loeb et al., 2009) |
| CMAP | merged analysis | pr | 2.5° x 2.5° | 1979-2013 | | Xie and Arkin (1997) |
| CRU | reanalysis | tas | 0.5° x 0.5° | 1901-2006 | | Harris et al. (2014) |
| ERA-Interim | reanalysis | tas, ua, va, pr, psl, LWP, IWP | 0.75° x 0.75° | 1979-2014 | | Dee et al. (2011) |
| HadISST | reanalysis | ts | 1° x 1° | 1870-2014 | | Rayner et al. (2003) |
| GPCP-1DD | satellite + gauge | pr | 1° x 1° | 1996-2010 | | Huffman et al. (2001) |
| GPCP-SG | satellite + gauge | pr | 2.5° x 2.5° | 1979-2013 | 0-2 mm day$^{-1}$ | Adler et al. (2003), Huffman and Bolvin (2012) |
| MODIS-L3 | satellite | clt, IWP | 1° x 1° | 2003-2014 | 15%, 125% | Platnick (2015) |
| NCEP | reanalysis | tas, ua, va | 2.5° x 2.5° | 1948-2012 | | Kalnay et al. (1996) |
| PATMOS-x | satellite | clt | 1° x 1° | 1982-2014 | | Heidinger et al. (2014) |
| SRB | satellite | lw_cre | 1° x 1° | 1984-2007 | | Zhang et al. (2009) |
| TRMM | satellite | pr | 0.25° x 0.25° | 1989-2011 | | Huffman et al. (2007) |
| UWisc | satellite | LWP | 1° x 1° | 1988-2007 | 15%-30% | O'Dell et al. (2008) |



**Table 4. Variables used.**

| Variable | Name | Unit | Comment |
| --- | --- | --- | --- |
| **clt** | Total cloud fraction | % | For the whole atmospheric column, as seen from the surface or the top of the atmosphere; includes both large-scale and convective clouds |
| **IWP** | Ice water path | $kg\ m^{-2}$ | |
| **LWP** | Liquid water path | $kg\ m^{-2}$ | |
| **lw_cre** | TOA longwave cloud radiative effect | $W\ m^{-2}$ | |
| **pr** | Precipitation | $kg\ m^{-2}\ s^{-1}$ | At surface; includes both liquid and solid phases from all types of clouds (both large-scale and convective) |
| **psl** | Air pressure at sea level | Pa | |
| **rlds (LWD)** | Surface downwelling longwave radiation | $W\ m^{-2}$ | At the surface |
| **rlut (OLR)** | TOA outgoing longwave radiation | $W\ m^{-2}$ | At the top of the atmosphere |
| **rsds (SWD)** | Surface downwelling shortwave radiation | $W\ m^{-2}$ | At the surface |
| **rsut (SWUP)** | TOA outgoing shortwave radiation | $W\ m^{-2}$ | At the top of the atmosphere |
| **tas** | Near-surface air temperature | K | |
| **sw_cre** | TOA shortwave cloud radiative effect | $W\ m^{-2}$ | |
| **ts** | Surface temperature | K | "skin" temperature (i.e., SST for open ocean) |
| **ua** | Eastward wind | $m\ s^{-1}$ | |
| **va** | Northward wind | $m\ s^{-1}$ | |