# Peer review of "Process-level improvements in CMIP5 models and their impact on tropical variability, Southern Ocean and monsoons"

_Earth System Dynamics, 2017_

## Referee Comment (RC1) · Anonymous Referee #1 · 8 Aug 2017

This manuscript by Dr. Lauer and colleagues examines the performance of four updated earth system models (ESMs) in simulating a selection of processes and climate phenomena for which ESMs have demonstrated a need for improvement. Specifically, this study focuses on ESMs that have been updated during the EMBRACE project. The authors focus on phenomena relating to the coupled tropical and Southern Ocean climate. Overall, the authors find that the updated models have improved over their CMIP5 counterparts in several respects, including simulation of mean tropical precipitation, synoptic-scale variability of tropical precipitation, equatorial sea surface temperatures (SSTs), and Southern Ocean cloud cover. The authors also note, however, that the simulation of some processes demonstrates little improvement relative to the

[Figure]

CMIP5 versions, and some process-level performance diagnoses are strongly constrained by observational data uncertainties.

Overall, I feel that this is a well written manuscript that provides useful diagnoses for the earth system modeling community. The study is well structured, and the analyses target important earth system processes and shed light on ESM development. The areas that could use improvement, in my opinion, relate to the synthesis of model performance. There are several areas where the message is not quite clear to me owing to conflicting evidence, observational uncertainty not clearly stated, or lack of specific evaluation criteria. I believe that this article will be a useful contribution to the literature if the authors can clarify several points raised below.

Specific Comments

1) The authors present the results under the presumption that all the ESMs being evaluated are improved (e.g. first line of the abstract). As I read this manuscript, I am left wondering if some of these ESMs are actually improved overall relative to their predecessor versions. Of the three fully coupled ESMs being evaluated, only EC-Earth seems to be consistently improved. If we use RMSE as an overall performance metric, HadGEM is approximately the same or worse in global surface temperature (Fig. 1), global precipitation (Fig. 2), South Asian monsoon climate (Figs. 3-4), and West African monsoon climate (Figs. 7-9 and 11). Similarly, MPI-ESM performance is approximately the same or worse in all of the same categories except global precipitation (Fig. 2) and African JJAS temperature (Fig. 8). With that in mind, are we confident that all the models are improved overall? The authors acknowledge that there are processes that have not shown improvement, but I believe that they could make some additional effort to synthesize the performance changes in the ESMs.

2) In several places, the authors claim that the models are "significantly" improved or have "significant" biases (p1, line 20; p8, line 1; p10, line 14; p13, line 2, etc.). It is difficult to see how significance is determined. There are no formal significance tests

performed. The analyses indicate large observational uncertainties, so it is difficult to determine by eye which model changes or biases are meaningful. I recommend that the authors specify the criteria used to determine meaningful improvements and biases. I also recommend omitting the word "significant" unless a formal significance test is indicated.

3) p. 4, lines 15-16: I do not understand what is meant by "medium range climate sensitivity." The phrase "medium range" typically refers to a period of time.

4) Section 3.1.1: This relates to my first comment, but the text does not seem to convey all the information evident in Fig. 1. First, I recommend that the authors indicate in the caption what the "bias" and "rmsd" numbers above each plot represent – presumably it is the global mean bias and root-mean-square deviation from ERA-Interim reanalysis. I also believe it is worth mentioning in the text that (1) there is large uncertainty in the near-surface air temperature climatology in reanalysis data (evident in the bottom right of Fig. 2), and (2) overall CNRM and EC-Earth show evidence of improved surface temperature climatology but HadGEM and MPI are not improved overall.

5) p. 8, lines 3-4: In addition to the general issue raised in my second comment, the bottom right figure seems to indicate that the amplitude of the tropical precipitation biases may be strongly dependent on the choice of observational dataset. How do the wet biases change with a CMAP reference climatology?

6) Fig. 3 and later figures: What are the "corr" values above each plot? Are those pattern correlations with the reference field? Again, these types of metrics should be described in the figure caption. In addition, the "Reference" label in the top left panel is a bit obstructive in some of the plots. This is especially true in Fig. 4, where the label obstructs the region of strongest horizontal temperature gradients.

7) Fig. 4: The top label in the second and fourth columns indicates "NCEP" but the figure caption indicates the CRU dataset.

8) Figs. 6 and 10: I don't understand what "yrs: inconsistent" means at the top of each panel.

9) p. 15, line 17: I don't follow what is meant by "natural and forced modes of variability" in that context.

10) Fig. 8: The top right panel indicates a CRU RMSE of 5.29. Is that a typo? If not, then the model changes illustrated in the figure are very difficult to interpret because they would be within the noise of observational uncertainty. The top right panel, however, does not look consistent with an RMSE that high.

11) Synoptic precipitation variability (pages 21-26): Overall, I found this section difficult to interpret. First, based on the plots and metrics presented in Fig. 11, it looks like all the EMBRACE models perform worse in simulating band-pass filtered precipitation variance over the illustrated domain – this point is not brought out in the text. (Also, I do not know what the "sahel" metric is above each plot – again, all metrics presented in the plots should be explained.) In addition, the authors argue that increasing atmospheric resolution improves the representation of precipitation variability. I am not sure how this evaluation is made. Figs. 11-13 indicate large observational uncertainty, depending on the use of TRMM or GPCP data. If GPCP is used as the baseline, it is not clear that increasing horizontal resolution leads to improvement. Are the authors assuming that TRMM is the more appropriate baseline? If so, why? And why do the EMBRACE models look much worse than their CMIP5 counterparts relative to TRMM (Fig. 11)?

This is a more minor issue, but I think it's worth mentioning that increasing horizontal resolution does not seem to improve deficiencies in the general pattern of elevated variance over Africa, which is too narrow and does not extend far enough south in the model (Fig. 12).

12) Figs. 17-19: I do not see any gray dots in the EC-EARTH plots. Is that because there is almost perfect overlap between EC-Earth and EC-Earth3?

13) Table 1 caption: I do not understand the use of the phrases "can be close" and "can include." It is trivial that the models can be close to the CMIP6 versions, but it is more important to know if they are.

Technical Corrections

1) p.1, line 15: "earth" -> "Earth" 2) p. 1, line 29: "over the last years" -> "in the recent past" or "over the past few years" 3) p. 2, line 3: "ITCZ" acronym should be expanded. 4) p.2, line 19: "is particularly focusing on" -> "has a particular focus on" 5) p. 12, line 1: "varies" -> "vary" 6) p. 15, line 24: "that one of is" -> "that is one of" 7) p. 19, line 2: "Further" -> "Farther" 8) p. 37, line 28: "further" -> "farther"
* * *

---

## Referee Comment (RC2) · Anonymous Referee #2 · 24 Aug 2017

Process-level improvements in CMIP5 models and their impact on tropical variability, Southern Ocean and monsoons. Lauer et al.

Submitted to Earth Systems Dynamics

The paper is a very thorough evaluation of several aspects of model performance between two generations of three climate models. There is a lot of useful evaluation material, with high quality analysis and presentation. I think the paper will be useful for the climate modelling community, where other modellers can take lessons from the model development that they can then apply to other models. Modellers may also benefit from seeing an application of the evaluation software, and see the utility for their

analysis. However, I think the usefulness of the evaluation for modellers and also a wider audience could be improved by drawing more links between the model performance and the differences between model versions (see major comments), and a few minor edits as well. I recommend publishing once reviews have been addressed

Scientific:

1. There needs to be more lessons about model development drawn out, so that the evaluation can be used in a constructive way by others. The analysis of resolution (3.2.3) is useful to identify the influence of this factor, what it provides and what it doesn't (e.g. improvement in moist processes but no improvement in AEW). But I feel the paper needs more links back to the cause of differences not just from resolution but in terms of model schemes and other model improvements. Comments and conclusion about what has led to improvements, what didn't, and what is still required would be useful. For example on page 37, line 16-31 when commenting on the remaining cloud and convection biases, then further comments about the model would be useful – e.g. what improvements were expected, what model components actually did contribute to model improvements, what these changes didn't achieve, and what are the remaining issues. Page 38 line 16-19 offers some insights into what caused some improvements in the Sahel, but the evidence supporting these claims is not laid out clearly, and is not taken to the level of modelling decisions (i.e. what caused the improvement in stratocumulus clouds, which components/s?).

2. For the benefit of the users of model outputs, I would like to see a discussion of where performance is 'good enough' for using the models for projections, and in what ways. I know this topic is difficult, but it is important context to judge the evaluation – at the moment, the reader is at a loss to know whether the old versions were not suitable for making particular climate projections but the new versions are suitable, or if both versions are still not useful, or both are good enough for a given purpose. This should be covered briefly throughout, in regard to each purpose, i.e. the simulation of mean rainfall to projections of mean regional rainfall, and so on. There are some

inferences made – e.g. page 21, line 6-10 suggests that models are not suitable to use for projections of changes to intra-seasonal variability in WAM rainfall in the old and the new version, and more like this would be useful.

3. Throughout – for evaluation, why use just 20 years? I guess this is the IPCC baseline so is common, but it is short enough to be strongly affected by variability such as mega-droughts. A longer period that still has satellite coverage (e.g. 1979-2016) would be better, then sub-periods within this could be also covered. If it is too expensive to run models for longer periods, then I understand this limitation is unavoidable and this should be mentioned. Also, figure captions need to note the time period in the figure captions (1986-2005).

4. The bias plots nicely show the differences at high values, but I think bias plots are more useful if they include a middle section of blank/white where the bias is negligible rather than having colours go to zero – this helps interpreting areas where biases are small and avoid over interpreting differences between positive and negative when in fact they are not meaningful (goes for all figures)

5. Pg 26 line 8-16 – this is one school of thought about ENSO and the tropical Pacific, but others would disagree – it is important to cover a range of ideas here. E.g. work by Felicity Graham: Graham et al. 2014 Effectiveness of the Bjerknes stability index in representing ocean dynamics (Climate Dynamics), and Graham et al. 2015 Reassessing conceptual models of ENSO (Journal of Climate)

6. Specify the version of NCEP used in each case – NCEP1 or 2?

7. Pg 2, Line 15 – why IPCC 2007 not 2013?

8. Fig 1 – I think an observations-based dataset should be used here, at least as the comparison panel, rather than two reanalyses. Also, are you reporting the difference between ERAint and NCEP as the observed uncertainty? If so, why not show panels only where the bias is larger than this observed uncertainty (blank out other regions)?

[Figure]

Could do this in other figures too.

9. Section 3.1.2 – absolute errors in rainfall appear higher where the mean is higher (i.e. the tropics), so reduce the appearance of biases in drier areas. The paper needs a figure showing proportional bias (%), even in additional material, to give some perspective on rainfall biases outside the tropics – for example the biases in Canada, Australia and Siberia look small and almost indistinguishable in the different panels, but important differences could be seen in a % bias plot. If % biases are extremely large due to extremely low rainfall, then these areas could be masked out or else identified and discussed.

10. Page 9, line 4 – I don't think projections will ever be 'accurate' in the sense they won't give a single, correct answer, so I think this word should be changed to "reliable", "robust" or similar – projections that give useful information but are not a single 'accurate' answer.

11. Figures 14-15 – I was expecting to see SST and rainfall bias map plots for the coupled versions (to see the shape of the warm pool, the extent of the cold tongue bias, the shape of the double ITCZ etc.) - one can see the temperature bias in Figure 1 somewhat, but it is not very clear. Perhaps a Pacific SST and rainfall bias map here or in additional material?

12. Page 38, line 20-24 – I think this conclusion is incomplete, yes the lack of improvement with finer resolution certainly indicates that there are problems at the coarse resolution and apparent good performance is probably related to compensating errors. But it also shows that these issues are not solved by finer resolution – thus indicating that either some critical threshold of resolution has not been reached (perhaps we need resolution somewhere <14 km) or else some other non-resolution factor is involved (e.g. parameterisations are not working well enough and the improvement to the cloud scheme didn't fix the problem). See major comment 1, I think this type of discussion and conclusion is needed more generally
Minor

1. Pg 2, Line 3 – ITCZ not defined on first usage in text

2. Pg 6, line 6 – Similar not similarly

3. Figure 2 – caption notes lower right panel is noted as data from CMIP, this should be CMAP

4. Figure 3 – the box with 'Reference' in it obscures an important high rainfall region in Nepal, suggest making it smaller and moving it (perhaps top left corner?), same in many other figures.

5. Page 15, line 7-8 – 1970s and 1980s don't need apostrophes,

6. Page 15, line 24 "that one of is the main" typo

7. Figure 10 caption and/or legend needs to explain the grey shading

8. Page 31, line 10 – a paper from 2010 can't show the 'current GCMs', note that this paper is about CMIP3

---

## Author Comment (AC1) · 20 Sep 2017

Below we address the comments of reviewer #1 and questions raised during the open discussion of the paper "Process-level improvements in CMIP5 models and their impact on tropical variability, Southern Ocean and monsoons". We would like to thank the reviewer for the time and effort reviewing the paper. We feel it has improved thanks to the constructive comments. We have listed all reviewer comments below and our answers are provided in blue.

[Figure]

Anonymous Referee #1

This manuscript by Dr. Lauer and colleagues examines the performance of four updated earth system models (ESMs) in simulating a selection of processes and climate phenomena for which ESMs have demonstrated a need for improvement. Specifically, this study focuses on ESMs that have been updated during the EMBRACE project. The authors focus on phenomena relating to the coupled tropical and Southern Ocean climate. Overall, the authors find that the updated models have improved over their CMIP5 counterparts in several respects, including simulation of mean tropical precipitation, synoptic-scale variability of tropical precipitation, equatorial sea surface temperatures (SSTs), and Southern Ocean cloud cover. The authors also note, however, that the simulation of some processes demonstrates little improvement relative to the CMIP5 versions, and some process-level performance diagnoses are strongly constrained by observational data uncertainties.

Overall, I feel that this is a well written manuscript that provides useful diagnoses for the earth system modeling community. The study is well structured, and the analyses target important earth system processes and shed light on ESM development. The areas that could use improvement, in my opinion, relate to the synthesis of model performance. There are several areas where the message is not quite clear to me owing to conflicting evidence, observational uncertainty not clearly stated, or lack of specific evaluation criteria. I believe that this article will be a useful contribution to the literature if the authors can clarify several points raised below.

Specific Comments

1) The authors present the results under the presumption that all the ESMs being evaluated are improved (e.g. first line of the abstract). As I read this manuscript,

I am left wondering if some of these ESMs are actually improved overall relative to their predecessor versions. Of the three fully coupled ESMs being evaluated, only EC-Earth seems to be consistently improved. If we use RMSE as an overall performance metric, HadGEM is approximately the same or worse in global surface temperature (Fig. 1), global precipitation (Fig. 2), South Asian monsoon climate (Figs. 3-4), and West African monsoon climate (Figs. 7-9 and 11). Similarly, MPI-ESM performance is approximately the same or worse in all of the same categories except global precipitation (Fig. 2) and African JJAS temperature (Fig. 8). With that in mind, are we confident that all the models are improved overall? The authors acknowledge that there are processes that have not shown improvement, but I believe that they could make some additional effort to synthesize the performance changes in the ESMs.

The word "improved" in the context of comparing the EMBRACE models to their CMIP5 predecessors was referring to the assumption that changes, modification and extensions applied to the models are either done in order to include more or more detailed processes or to improve the performance or skill level of the models. We agree with the reviewer that the word "improved" might be mistaken for referring to the model performance only and will replace "improved" with "updated" in the revised manuscript.

The reviewer is right that, when looking at global average RMSE values, only EC-EARTH has improved consistently. As we think that such global averages are only a starting point for a model evaluation, we looked at several regions and processes in more detail. As stated in the text, the models do show some improvements (in terms of better performance) in some regions, while they perform similarly or sometime even worse in other regions. It is therefore hard to define whether or not the models' overall performance is improved as this strongly depends on the focus of interest / region / process.

[Figure]

In order to address the legitimate criticisms of the reviewer, we will put more emphasis on synthesizing the performance changes by adding more discussion to each section in the revised manuscript on whether we think that the EMBRACE models show an improved performance compared to their CMIP5 predecessors. We will also make it clearer that the EMBRACE model versions were prototypes, not yet fully tuned, calibrated or developed with one aim of this paper to document the long and sometimes difficult pathway of model development and the challenges of improving large model biases.

2) In several places, the authors claim that the models are "significantly" improved or have "significant" biases (p1, line 20; p8, line 1; p10, line 14; p13, line 2, etc.). It is difficult to see how significance is determined. There are no formal significance tests performed. The analyses indicate large observational uncertainties, so it is difficult to determine by eye which model changes or biases are meaningful. I recommend that the authors specify the criteria used to determine meaningful improvements and biases. I also recommend omitting the word "significant" unless a formal significance test is indicated.

We agree with the reviewer that the word "significant" in this context is potentially misleading and will rephrase these sentences in the revised version.

3) p. 4, lines 15-16: I do not understand what is meant by "medium range climate sensitivity." The phrase "medium range" typically refers to a period of time.

"Medium range" was referring to the climate sensitivity of the new MPI-ESM model being in the middle of the range of climate sensitivities spanned by the ensemble of

CMIP models. The sentence will be clarified and rephrased.

4) Section 3.1.1: This relates to my first comment, but the text does not seem to convey all the information evident in Fig. 1. First, I recommend that the authors indicate in the caption what the "bias" and "rmsd" numbers above each plot represent – presumably it is the global mean bias and root-mean-square deviation from ERA-Interim reanalysis. I also believe it is worth mentioning in the text that (1) there is large uncertainty in the near-surface air temperature climatology in reanalysis data (evident in the bottom right of Fig. 2), and (2) overall CNRM and EC-Earth show evidence of improved surface temperature climatology but HadGEM and MPI are not improved overall.

The reviewer is right, the numbers given above each panel in figure 1 are global average values for the deviation of the model results from the ERA-Interim reanalysis. This will be clarified in the revised captions. The regions with particularly large uncertainties in the ERA-Interim data will be discussed in more detail and a panel showing Met Office Hadley Centre observations "HadCRUT" as an alternative reference will be added. We agree with the reviewer that the MPI model performance has not changed much (see p. 5, l. 14-15 and p. 6, l. 4-5). The inferior performance of the EMBRACE version of the HadGEM model in reproducing the ERA-Interim surface temperatures is stated in the text (p. 5, l. 30-31): "In these regions, the near-surface temperature biases in the EMBRACE version [of HadGEM] are up to 2°C larger than in the predecessor version.".

5) p. 8, lines 3-4: In addition to the general issue raised in my second comment, the bottom right figure seems to indicate that the amplitude of the tropical precipitation biases may be strongly dependent on the choice of observational dataset. How do the wet biases change with a CMAP reference climatology?

When using the CMAP dataset as reference, the model biases in the tropics are typically between 0.5-1 mm day$^{-1}$ smaller, in the mid-latitude storm track regions, the model bias changes sign from too dry (GPCP) to too wet (CMAP). The reviewer has a valid point and we will include the difference plots shown in figure 2 using CMAP as reference dataset in the supplementary material (that will be created along with the revised version).

6) Fig. 3 and later figures: What are the "corr" values above each plot? Are those pattern correlations with the reference field? Again, these types of metrics should be described in the figure caption. In addition, the "Reference" label in the top left panel is a bit obstructive in some of the plots. This is especially true in Fig. 4, where the label obstructs the region of strongest horizontal temperature gradients.

The reviewer is right, "corr" refers to the linear pattern correlation coefficients with the reference dataset. This will be added to the respective figure captions. The "reference" label will be reduced in size in the revised figures.

7) Fig. 4: The top label in the second and fourth columns indicates "NCEP" but the figure caption indicates the CRU dataset.

The figure is indeed showing results from NCEP. The figure caption will be corrected. Thanks for spotting this.

8) Figs. 6 and 10: I don't understand what "yrs: inconsistent" means at the top of each panel.

The labels "yrs: inconsistent" in figures 6 and 10 refer to the fact that the different datasets do not cover the same years when averaged (as can be seen from table 3). The label will be removed in the revised figures to avoid confusion and a reference to table 3 will be added to the figure caption.

9) p. 15, line 17: I don't follow what is meant by "natural and forced modes of variability" in that context.

This sentence refers to natural variability and a forced response to increased greenhouse gases driving precipitation changes. This will be clarified by rephrasing the sentence.

10) Fig. 8: The top right panel indicates a CRU RMSE of 5.29. Is that a typo? If not, then the model changes illustrated in the figure are very difficult to interpret because they would be within the noise of observational uncertainty. The top right panel, however, does not look consistent with an RMSE that high.

The RMSE given above the CRU panel is not matching the plot shown by mistake. Thanks for spotting this. This will be corrected in the revised version of the figure. The correct RMSE of the CRU dataset compared with ERA-Interim is 1.3 K, which is smaller than the RMSE of the models analyzed.

11) Synoptic precipitation variability (pages 21-26): Overall, I found this section difficult to interpret. First, based on the plots and metrics presented in Fig. 11, it looks like all the EMBRACE models perform worse in simulating band-pass filtered precipitation variance over the illustrated domain – this point is not brought out in the text. (Also, I do not know what the "sahel" metric is above each plot – again, all metrics presented

in the plots should be explained.) In addition, the authors argue that increasing atmospheric resolution improves the representation of precipitation variability. I am not sure how this evaluation is made. Figs. 11-13 indicate large observational uncertainty, depending on the use of TRMM or GPCP data. If GPCP is used as the baseline, it is not clear that increasing horizontal resolution leads to improvement. Are the authors assuming that TRMM is the more appropriate baseline? If so, why? And why do the EMBRACE models look much worse than their CMIP5 counterparts relative to TRMM (Fig. 11)?

This is a more minor issue, but I think it's worth mentioning that increasing horizontal resolution does not seem to improve deficiencies in the general pattern of elevated variance over Africa, which is too narrow and does not extend far enough south in the model (Fig. 12).

We agree with the reviewer that the EMBRACE models show less variability in the band-pass filtered daily precipitation fields than their CMIP5 counterparts. This will be explicitly mentioned in the revised version. The metrics "sahel" given above the individual panels in figure 11 is the band-pass filtered precipitation variance averaged over the rectangular region "Sahel" defined as 10°W-10°E, 10°N-20°N. This will be added to the caption of figure 11.

We also agree with the reviewer that the precipitation observations are subject to large uncertainties, in particular in the region investigated as there is only a sparse coverage with rain gauges. We use TRMM satellite observations as our reference dataset as the data have the highest horizontal resolution (0.25°x0.25°) of the three precipitation climatologies used (GPCP, TRMM, CMAP). We found that when regridding the TRMM data to a coarser grid such as 1°x1°, the band-pass filtered variability decreases. This can be seen when comparing the panels "TRMM" in figures 11 and 12: the TRMM data in figure 11 have been regridded to 1°x1° while the data are shown at full resolution (0.25°x0.25°) in figure 12 (note the different color scales). In order to improve the comparability, we will regrid all data (models and observations) to a common 2.5°x2.5°

grid when preparing the revised version of figure 11.

The evaluation performed here aims mainly at comparing CMIP5 version and EM-BRACE version rather than comparing the model results with the reference dataset. This will be clarified in the revised version of the manuscript by adding an additional discussion.

We will also add a brief discussion of the fact that the band of elevated variance is too narrow in the models compared with both observational datasets.

12) Figs. 17-19: I do not see any gray dots in the EC-EARTH plots. Is that because there is almost perfect overlap between EC-Earth and EC-Earth3?

Unfortunately, no radiation data from EC-EARTH are available, i.e. only the fractional occurrence of cloud cover can be shown for EC-EARTH (middle rows in figures 17-19). This will be added to the figure caption.

13) Table 1 caption: I do not understand the use of the phrases "can be close" and "can include." It is trivial that the models can be close to the CMIP6 versions, but it is more important to know if they are.

The caption of table 1 will be shortened to "List of models analyzed."

Technical Corrections

1) p.1, line 15: "earth" → "Earth"

2) p. 1, line 29: "over the last years" → "in the recent past" or "over the past few years"

3) p. 2, line 3: "ITCZ" acronym should be expanded.

4) p.2, line 19: "is particularly focusing on" → "has a particular focus on"

5) p. 12, line 1: "varies" → "vary"

6) p. 15, line 24: "that one of is" → "that is one of"

7) p. 19, line 2: "Further" → "Farther"

8) p. 37, line 28: "further" → "farther"

All technical corrections will be applied as suggested.

---

## Author Comment (AC2) · 20 Sep 2017

Below we reply to the anonymous referee #2's comments and questions on our ESDD manuscript "Process-level improvements in CMIP5 models and their impact on tropical variability, Southern Ocean and monsoons". We would like to thank the reviewer for the constructive comments helping us to improve the paper. We have listed all reviewer comments below and answers are provided in blue.

[Figure]

Anonymous Referee #2

The paper is a very thorough evaluation of several aspects of model performance between two generations of three climate models. There is a lot of useful evaluation material, with high quality analysis and presentation. I think the paper will be useful for the climate modelling community, where other modellers can take lessons from the model development that they can then apply to other models. Modellers may also benefit from seeing an application of the evaluation software, and see the utility for their analysis. However, I think the usefulness of the evaluation for modellers and also a wider audience could be improved by drawing more links between the model performance and the differences between model versions (see major comments), and a few minor edits as well. I recommend publishing once reviews have been addressed

Scientific:

1. There needs to be more lessons about model development drawn out, so that the evaluation can be used in a constructive way by others. The analysis of resolution (3.2.3) is useful to identify the influence of this factor, what it provides and what it doesn't (e.g. improvement in moist processes but no improvement in AEW). But I feel the paper needs more links back to the cause of differences not just from resolution but in terms of model schemes and other model improvements. Comments and conclusion about what has led to improvements, what didn't, and what is still required would be useful. For example on page 37, line 16-31 when commenting on the remaining cloud and convection biases, then further comments about the model would be useful – e.g. what improvements were expected, what model components actually did contribute to model improvements, what these changes didn't achieve, and what are the remaining issues. Page 38 line 16-19 offers some insights into what caused some improvements in the Sahel, but the evidence supporting these claims is not laid

out clearly, and is not taken to the level of modelling decisions (i.e. what caused the improvement in stratocumulus clouds, which components/s?).

The model simulations evaluated here were performed within the project EMBRACE, which was aiming at improving the models in preparation for CMIP6. Besides the targets for improvements in the representation of key variables and processes discussed in the manuscript, the model development also included biogeochemical mixing in the Southern ocean, soil hydrology, the carbon cycle, and a more realistic treatment of climate-vegetation interaction.
Model simulations with one individual component changed at a time suitable for an evaluation and comparison with their CMIP5 counterparts are not available because of computational constraints. As the new models are compared with their CMIP5 counterparts, only variables and derived quantities also available from CMIP5 can be included in the evaluation. This makes identification of the exact causes of differences between the CMIP5 and EMBRACE models quite challenging. In some cases possible reasons for the differences seen can be given but not in all cases. This also makes it very difficult to give clear advice for future model development. We will, however, elaborate more on possible reasons for model improvements and non-improvements where possible.

2. For the benefit of the users of model outputs, I would like to see a discussion of where performance is 'good enough' for using the models for projections, and in what ways. I know this topic is difficult, but it is important context to judge the evaluation – at the moment, the reader is at a loss to know whether the old versions were not suitable for making particular climate projections but the new versions are suitable, or if both versions are still not useful, or both are good enough for a given purpose. This should be covered briefly throughout, in regard to each purpose, i.e. the simulation of mean rainfall to projections of mean regional rainfall, and so on. There are some

inferences made – e.g. page 21, line 6-10 suggests that models are not suitable to use for projections of changes to intra-seasonal variability in WAM rainfall in the old and the new version, and more like this would be useful.

The reviewer has a very good point. The usability of model results for supporting policy relevant decisions is an important aspect for users of the model data. The question of "how good is good enough" for using model results for any kind of application depends strongly on the process of interest. This includes details such as geographical region, simulated quantity, natural variability, time-scales and time range or metric (e.g. mean, extreme values, probability density function, etc.). We feel that statements on the usability or usefulness of the model results are beyond the scope of this study because of the mentioned complexity of the problem and also because of the high sensitivity of this topic. Similar to the model evaluation chapter (chapter 9) of the fifth Assessment Report of the Intergovernmental Panel on Climate Change (IPCC-AR5), we regard "the need for climate models to represent the observed behaviour of past climate" only as a "necessary condition to be considered a viable tool for future projections". This does not answer the "much more difficult question of determining how well a model must agree with observations before projections made with it can be deemed reliable." (IPCC-AR5)
We will, however, add a brief general discussion of this topic to the introduction of the revised version noting that even a "decent" fit does not necessarily guarantee a correct model behaviour in future (and changing) climate predictions, which is one of the reasons why ensemble based methods are used.

3. Throughout – for evaluation, why use just 20 years? I guess this is the IPCC baseline so is common, but it is short enough to be strongly affected by variability such as megadroughts. A longer period that still has satellite coverage (e.g. 1979-2016) would be better, then sub-periods within this could be also covered. If it is too expensive to
run models for longer periods, then I understand this limitation is unavoidable and this should be mentioned. Also, figure captions need to note the time period in the figure captions (1986-2005).

The CMIP5 simulations analyzed (AMIP, historical) typically only cover the time period up to the year 2005. In order to maximize the comparability of the EMBRACE and CMIP5 simulations and with results from the model evaluation chapter (chapter 9) of IPCC-AR5, we decided to use the time period 1986-2005 as a common denominator even though sometimes more years are available. The figure captions will be updated to include the time periods shown.

4. The bias plots nicely show the differences at high values, but I think bias plots are more useful if they include a middle section of blank/white where the bias is negligible rather than having colours go to zero – this helps interpreting areas where biases are small and avoid over interpreting differences between positive and negative when in fact they are not meaningful (goes for all figures)

The color scales are similar to the ones used in chapter 9 of IPCC-AR5. We would therefore prefer to keep the color scales as they are as this allows for an easier comparison with the multi-model mean results shown in the IPCC-AR5. This will be clarified in the revised manuscript.

5. Pg 26 line 8-16 – this is one school of thought about ENSO and the tropical Pacific, but others would disagree – it is important to cover a range of ideas here. E.g. work by Felicity Graham: Graham et al. 2014 Effectiveness of the Bjerknes stability index in representing ocean dynamics (Climate Dynamics), and Graham et al. 2015 Reassessing conceptual models of ENSO (Journal of Climate)

We will add the alternative conceptual model for ENSO suggested by the reviewer to the introduction of section 3.3 (coupled tropical ocean climate).

6. Specify the version of NCEP used in each case – NCEP1 or 2?

All NCEP data used are NCEP 1 data (Kalnay et al., 1996). This will be clarified in the revised version.

7. Pg 2, Line 15 – why IPCC 2007 not 2013?

We will add IPCC 2013 as a reference.

8. Fig 1 – I think an observations-based dataset should be used here, at least as the comparison panel, rather than two reanalyses. Also, are you reporting the difference between ERAint and NCEP as the observed uncertainty? If so, why not show panels only where the bias is larger than this observed uncertainty (blank out other regions)? Could do this in other figures too.

We are showing differences compared to ERA-Interim to allow for easier comparison with the CMIP5 multi-model mean shown in the IPCC-AR5. The same is true for the color scale. We would therefore prefer to keep ERA-Interim as the reference dataset and the color scales as they are. The comparison with NCEP is shown to highlight the regions with particularly large uncertainties in the reanalyses. In order to strengthen this point, we will add a panel showing the Met Office Hadley Centre observations "HadCRUT" to the figure.

[Figure]

9. Section 3.1.2 – absolute errors in rainfall appear higher where the mean is higher (i.e. the tropics), so reduce the appearance of biases in drier areas. The paper needs a figure showing proportional bias (%), even in additional material, to give some perspective on rainfall biases outside the tropics – for example the biases in Canada, Australia and Siberia look small and almost indistinguishable in the different panels, but important differences could be seen in a % bias plot. If % biases are extremely large due to extremely low rainfall, then these areas could be masked out or else identified and discussed.

We will add a figure showing the relative bias in precipitation to the supplementary material (that will be newly created).

10. Page 9, line 4 – I don't think projections will ever be 'accurate' in the sense they won't give a single, correct answer, so I think this word should be changed to "reliable", "robust" or similar – projections that give useful information but are not a single 'accurate' answer.

We will replace accurate by "reliable" as suggested.

11. Figures 14-15 – I was expecting to see SST and rainfall bias map plots for the coupled versions (to see the shape of the warm pool, the extent of the cold tongue bias, the shape of the double ITCZ etc.) – one can see the temperature bias in Figure 1 somewhat, but it is not very clear. Perhaps a Pacific SST and rainfall bias map here or in additional material?

[Figure]

Following the suggestion of the reviewer, we will add SST and rainfall bias maps zoomed in over the Pacific to the supplementary material (that will be newly created).

12. Page 38, line 20-24 – I think this conclusion is incomplete, yes the lack of improvement with finer resolution certainly indicates that there are problems at the coarse resolution and apparent good performance is probably related to compensating errors. But it also shows that these issues are not solved by finer resolution – thus indicating that either some critical threshold of resolution has not been reached (perhaps we need resolution somewhere <14 km) or else some other non-resolution factor is involved (e.g. parameterisations are not working well enough and the improvement to the cloud scheme didn't fix the problem). See major comment 1, I think this type of discussion and conclusion is needed more generally

We agree with the reviewer that the results suggest that there are most likely also factors other than horizontal resolution involved. We will extend the discussion in the revised manuscript to include this conclusion.

Minor

1. Pg 2, Line 3 – ITCZ not defined on first usage in text

2. Pg 6, line 6 – Similar not similarly

3. Figure 2 – caption notes lower right panel is noted as data from CMIP, this should be CMAP

4. Figure 3 – the box with 'Reference' in it obscures an important high rainfall region in Nepal, suggest making it smaller and moving it (perhaps top left corner?), same in many other figures.

5. Page 15, line 7-8 – 1970s and 1980s don't need apostrophes,

6. Page 15, line 24 "that one of is the main" typo

7. Figure 10 caption and/or legend needs to explain the grey shading

8. Page 31, line 10 – a paper from 2010 can't show the 'current GCMs', note that this paper is about CMIP3

All minor suggestions / corrections will be applied as suggested.

---

## Author Response (AR1)

**Response to Reviewers**
**Earth System Dynamics**

Manuscript:       esd-2017-61
Title:             Process-level improvements in CMIP5 models and their impact on tropical variability, Southern Ocean and monsoons

Authors:           Axel Lauer, Colin Jones, Veronika Eyring, Martin Evaldsson, Stefan Hagemann, Jarmo Mäkelä, Gill Martin, Romain Roehrig, and Shiyu Wang

*The original reviewer comments are given in blue. If not otherwise noted, all page and line numbers refer to the "track changes" version of the revised manuscript.*

**Anonymous Referee #1**

This manuscript by Dr. Lauer and colleagues examines the performance of four updated earth system models (ESMs) in simulating a selection of processes and climate phenomena for which ESMs have demonstrated a need for improvement. Specifically, this study focuses on ESMs that have been updated during the EMBRACE project. The authors focus on phenomena relating to the coupled tropical and Southern Ocean climate. Overall, the authors find that the updated models have improved over their CMIP5 counterparts in several respects, including simulation of mean tropical precipitation, synoptic-scale variability of tropical precipitation, equatorial sea surface temperatures (SSTs), and Southern Ocean cloud cover. The authors also note, however, that the simulation of some processes demonstrates little improvement relative to the CMIP5 versions, and some process-level performance diagnoses are strongly constrained by observational data uncertainties.
Overall, I feel that this is a well written manuscript that provides useful diagnoses for the earth system modeling community. The study is well structured, and the analyses target important earth system processes and shed light on ESM development. The areas that could use improvement, in my opinion, relate to the synthesis of model performance. There are several areas where the message is not quite clear to me owing to conflicting evidence, observational uncertainty not clearly stated, or lack of specific evaluation criteria. I believe that this article will be a useful contribution to the literature if the authors can clarify several points raised below.

We thank Reviewer #1 for providing helpful comments to improve the manuscript.

Specific Comments

1) The authors present the results under the presumption that all the ESMs being evaluated are improved (e.g. first line of the abstract). As I read this manuscript, I am left wondering if some of these ESMs are actually improved overall relative to their predecessor versions. Of the three fully coupled ESMs being evaluated, only EC-Earth seems to be consistently improved. If we use RMSE as an overall performance metric, HadGEM is approximately the same or worse in global surface temperature (Fig. 1), global precipitation (Fig. 2), South Asian monsoon climate (Figs. 3-4), and West African monsoon climate (Figs. 7-9 and 11). Similarly, MPI-ESM performance is approximately the same or worse in all of the same categories except global precipitation (Fig. 2) and African JJAS temperature (Fig. 8). With that in mind, are we confident that all the models are improved overall? The authors acknowledge that there are processes that have not shown improvement, but I believe that they could make some additional effort to synthesize the performance changes in the ESMs.

The word "improved" in the context of comparing the EMBRACE models to their CMIP5 predecessors was referring to the assumption that changes, modification and extensions applied to the models are either done in order to include more or more detailed processes or to improve the performance or skill level of the models. We agree with the reviewer that the word "improved" might be mistaken for referring to the model performance and replaced "improved" with "updated" throughout the revised manuscript.
The reviewer is right that, when looking at global average RMSE values, only EC-Earth has improved consistently. As we think that such (global) averages are only a starting point for a model evaluation, we looked at several regions and processes in more detail. As stated in the text, the models do show some improvements (in terms of better performance) in some regions, while they perform similarly or sometime even worse in other regions. It is therefore hard to define whether or not the models' overall performance is improved as this strongly depends on the focus of interest / region / process.

In order to address the legitimate criticisms of the reviewer, we put more emphasis on synthesizing the performance changes by extending many of the sections with brief discussions on whether we think that the EMBRACE models show an improved performance compared to their CMIP5 predecessors (p.6/l.31-p.7/l.2, p.18/l.3-10, p.30/l.1-6, p.36/l.14-15).

2) In several places, the authors claim that the models are "significantly" improved or have "significant" biases (p1, line 20; p8, line 1; p10, line 14; p13, line 2, etc.). It is difficult to see how significance is determined. There are no formal significance tests performed. The analyses indicate large observational uncertainties, so it is difficult to determine by eye which model changes or biases are meaningful. I recommend that the authors specify the criteria used to determine meaningful improvements and biases. I also recommend omitting the word "significant" unless a formal significance test is indicated.

We agree with the reviewer that the word "significant" in this context is potentially misleading and rephrased these sentences in the revised version.

3) p. 4, lines 15-16: I do not understand what is meant by "medium range climate sensitivity." The phrase "medium range" typically refers to a period of time.

"Medium range" was referring to the climate sensitivity of the new MPI-ESM model being in the middle of the range of climate sensitivities spanned by the ensemble of CMIP models. The sentence has been clarified and rephrased (p.4/l.27-29).

4) Section 3.1.1: This relates to my first comment, but the text does not seem to convey all the information evident in Fig. 1. First, I recommend that the authors indicate in the caption what the "bias" and "rmsd" numbers above each plot represent – presumably it is the global mean bias and root-mean-square deviation from ERA-Interim reanalysis. I also believe it is worth mentioning in the text that (1) there is large uncertainty in the near-surface air temperature climatology in reanalysis data (evident in the bottom right of Fig. 2), and (2) overall CNRM and EC-Earth show evidence of improved surface temperature climatology but HadGEM and MPI are not improved overall.

The reviewer is right, the numbers given above each panel in Fig. 1 are global average values for the deviation of the model results from the ERA-Interim reanalysis. This has been clarified in the revised captions (Figs. 1, 2). The regions with particularly large uncertainties in the ERA-Interim data are now discussed in more detail (p.6/l.27-30) and a panel showing the Climate Research Unit (CRU) dataset as an alternative reference has been added to Fig. 1.
We agree with the reviewer that the MPI model performance has not changed much (p.5/l.27-29 and p.6/l.20-21). The inferior performance of the EMBRACE version of the HadGEM model in reproducing the ERA-Interim surface temperatures is already stated in the text (p.6/l.14-15).

5) p. 8, lines 3-4: In addition to the general issue raised in my second comment, the bottom right figure seems to indicate that the amplitude of the tropical precipitation biases may be strongly dependent on the choice of observational dataset. How do the wet biases change with a CMAP reference climatology?

When using the CMAP dataset as reference, the model biases in the tropics are typically between 0.5-1 mm day$^{-1}$ smaller. In the mid-latitude storm track regions, the model bias changes sign from too dry (GPCP) to too wet (CMAP). The reduction in the tropical rain bias in the EMBRACE models discussed as the main improvement, however, holds when using CMAP as a reference. This has been added to section 3.1.2 (p.10/l.30-33). As suggested by the reviewer, we also included the difference plots shown in Fig. 2 using CMAP as reference dataset in the supplementary material (Fig. S2).

6) Fig. 3 and later figures: What are the "corr" values above each plot? Are those pattern correlations with the reference field? Again, these types of metrics should be described in the figure caption. In addition, the "Reference" label in the top left panel is a bit obstructive in some of the plots. This is especially true in Fig. 4, where the label obstructs the region of strongest horizontal temperature gradients.

The reviewer is right, "corr" refers to the linear pattern correlation coefficients with the reference dataset. This has been added to the respective figure captions (Figs. 1, 2, 3, 4, 5, 7, 8, 9). The "reference" label has been reduced in size in the revised figures (Figs. 3, 4, 5, 7, 8, 9, 11).

7) Fig. 4: The top label in the second and fourth columns indicates "NCEP" but the figure caption indicates the CRU dataset.

The figure is indeed showing results from NCEP. The caption of Fig. 4 has been corrected. Thanks for spotting this.

8) Figs. 6 and 10: I don't understand what "yrs: inconsistent" means at the top of each panel.

The labels "yrs: inconsistent" in Figs. 6 and 10 refer to the fact that the different datasets do not cover the same years (as listed in Tab. 3). The labels have been removed in the revised figures (Figs. 6, 10) and the years averaged have been added to the figure captions instead.

9) p. 15, line 17: I don't follow what is meant by "natural and forced modes of variability" in that context.

This sentence refers to natural variability and a forced response to increased greenhouse gases driving precipitation changes. This has been clarified by rephrasing the sentence (p.24/l.16-19).

10) Fig. 8: The top right panel indicates a CRU RMSE of 5.29. Is that a typo? If not, then the model changes illustrated in the figure are very difficult to interpret because they would be within the noise of observational uncertainty. The top right panel, however, does not look consistent with an RMSE that high.

The RMSE given above the CRU panel did not match the plot shown by mistake. Thanks for spotting this. This has been corrected in the revised version of Fig. 8. The correct RMSE of the CRU dataset compared with ERA-Interim is 1.3 K, which is smaller than the RMSE of the models analyzed.

11) Synoptic precipitation variability (pages 21-26): Overall, I found this section difficult to interpret. First, based on the plots and metrics presented in Fig. 11, it looks like all the EMBRACE models perform worse in simulating band-pass filtered precipitation variance over the illustrated domain – this point is not brought out in the text. (Also, I do not know what the "sahel" metric is above each plot – again, all metrics presented in the plots should be explained.) In addition, the authors argue that increasing atmospheric resolution improves the representation of precipitation variability. I am not sure how this evaluation is made. Figs. 11-13 indicate large observational uncertainty, depending on the use of TRMM or GPCP data. If GPCP is used as the baseline, it is not clear that increasing horizontal resolution leads to improvement. Are the authors assuming that TRMM is the more appropriate baseline? If so, why? And why do the EMBRACE models look much worse than their CMIP5 counterparts relative to TRMM (Fig. 11)?

This is a more minor issue, but I think it's worth mentioning that increasing horizontal resolution does not seem to improve deficiencies in the general pattern of elevated variance over Africa, which is too narrow and does not extend far enough south in the model (Fig. 12).

We deliberately did not contrast the performance of the EMBRACE and CMIP5 coupled models in terms of precipitation variability. Fig. 11 is rather just used to emphasize that coupled GCMs with resolutions typical of CMIP5 models simulate precipitation variability in the SAM region poorly (when evaluated against TRMM data which we think give more realistic precipitation variability than GPCP-1DD as explained below). This figure was motivation for the "EC-Earth resolution figures" (Figs. 12, 13).

The metrics "sahel" given above the individual panels in Fig. 11 is the 3-10 day band-pass filtered precipitation variance averaged over the rectangular region "Sahel" here defined as 10°W-10°E, 10°N-20°N. This has been added to the caption of Fig. 11.

We agree with the reviewer that the precipitation observations are subject to large uncertainties, in particular in the region investigated as there is only a sparse coverage with rain gauges. The base TRMM observational data are at 0.25° spatial resolution and 3-hourly temporal resolution, whereas the GPCP-1DD data are at a spatial resolution of 1° and the highest temporal resolution is daily mean values. Due to the low spatial and temporal resolution of GPCP-1DD and the high (time and space) variability of (actual) convective precipitation over both India and Africa, GPCP-1DD will mix (in both space and time) areas of active precipitation with cloudy rain free and cloud-free regions/periods. Hence even if the two data sets were based on the same underlying retrieval mechanisms, solely due the higher time/space resolution we would expect that TRMM samples more accurately the true (time and space) variability of precipitation than GPCP-1DD. Furthermore, due to the non-Gaussian (space/time) distribution of convective precipitation we expect the sampled variability to increase significantly as resolution increases, as is seen in the TRMM data relative to the GPCP-1DD data. This increase in (observed) space/time precipitation variability occurs with only a relatively small change in the time mean (climatological) precipitation. This has been added to the revised manuscript (p.36/l.6-11).

As model resolution is increased in the EC-Earth simulations we see a similar increase in space/time precipitation variability (also with little change in the simulated time mean climatological precipitation across the EC-Earth model versions) suggesting EC-Earth captures precipitation variability more realistically as horizontal resolution is increased. This increase in variability in the EC-Earth simulations stops around T511 (~40km), with little continued increase as horizontal resolution is increased further. While we do not have gridded observational data to assess whether observed precipitation variability would continue to increase beyond the resolution of TRMM, we contend it likely would, asymptoting near the typical resolution of deep convective systems over these regions (kilometer scale). Hence the increase in variability towards that seen in TRMM as EC-Earth tends towards 0.25° is a good model result, but the lack of continued variability increase as resolution increases further is (likely) a model deficiency. Other factors such as such as, for instance, deficiencies in the cloud and precipitation parameterizations are also expected to contribute. This has been added to the revised manuscript (p.42/l.7-9).

We cannot comment on the specifics of why the EMBRACE models do relatively poorly for this metric in the SAM region (this has been added to the manuscript, p.36/l.14-15), but highlight that the EMBRACE models are in the development phase as groups implement improvements (developed in EMBRACE and elsewhere) to their fully coupled models. Ultimately, however, the path of model development is complex and it is rarely possible to achieve even a final model configuration that provides improvements in all aspects of the climate simulation.

12) Figs. 17-19: I do not see any gray dots in the EC-EARTH plots. Is that because there is almost perfect overlap between EC-Earth and EC-Earth3?

Unfortunately, no radiation data from EC-EARTH are available, i.e. only the fractional occurrence of cloud cover can be shown for EC-Earth (middle rows in Figs. 17, 18, 19). This has been added to the caption of Fig. 17.

13) Table 1 caption: I do not understand the use of the phrases "can be close" and "can include." It is trivial that the models can be close to the CMIP6 versions, but it is more important to know if they are.

The caption of Tab. 1 has been shortened to "List of models analyzed."

Technical Corrections

1) p.1, line 15: "earth" -> "Earth"

2) p. 1, line 29: "over the last years" -> "in the recent past" or "over the past few years"

3) p. 2, line 3: "ITCZ" acronym should be expanded.

4) p.2, line 19: "is particularly focusing on" -> "has a particular focus on"

5) p. 12, line 1: "varies" -> "vary"

6) p. 15, line 24: "that one of is" -> "that is one of"

7) p. 19, line 2: "Further" -> "Farther"

8) p. 37, line 28: "further" -> "farther"

All technical corrections have been applied as suggested.

**Anonymous Referee #2**

The paper is a very thorough evaluation of several aspects of model performance between two generations of three climate models. There is a lot of useful evaluation material, with high quality analysis and presentation. I think the paper will be useful for the climate modelling community, where other modellers can take lessons from the model development that they can then apply to other models. Modellers may also benefit from seeing an application of the evaluation software, and see the utility for their analysis. However, I think the usefulness of the evaluation for modellers and also a wider audience could be improved by drawing more links between the model performance and the differences between model versions (see major comments), and a few minor edits as well. I recommend publishing once reviews have been addressed

We would also like to thank Reviewer #2 for helping us to improve the manuscript.

Scientific:

1. There needs to be more lessons about model development drawn out, so that the evaluation can be used in a constructive way by others. The analysis of resolution (3.2.3) is useful to identify the influence of this factor, what it provides and what it doesn't (e.g. improvement in moist processes but no improvement in AEW). But I feel the paper needs more links back to the cause of differences not just from resolution but in terms of model schemes and other model improvements. Comments and conclusion about what has led to improvements, what didn't, and what is still required would be useful. For example on page 37, line 16-31 when commenting on the remaining cloud and convection biases, then further comments about the model would be useful – e.g. what improvements were expected, what model components actually did contribute to model improvements, what these changes didn't achieve, and what are the remaining issues. Page 38 line 16-19 offers some insights into what caused some improvements in the Sahel, but the evidence supporting these claims is not laid out clearly, and is not taken to the level of modelling decisions (i.e. what caused the improvement in stratocumulus clouds, which components/s?).

The model simulations evaluated here were performed within the project EMBRACE, which was aiming at updating/improving the models in preparation for CMIP6. Besides the targets for improvements in the representation of key variables and processes discussed in the manuscript, the model development also included biogeochemical mixing in the Southern Ocean, soil hydrology, the carbon cycle, and a more realistic treatment of climate-vegetation interaction.

Model simulations with one individual component changed at a time suitable for an evaluation and comparison with their CMIP5 counterparts are not available because of computational constraints. As the new models are compared with their CMIP5 counterparts, only variables and derived quantities also available from CMIP5 can be included in the evaluation. This makes identification of the exact causes of differences between the CMIP5 and EMBRACE models quite challenging. In some cases possible reasons for the differences found can be given but not in all cases. This also makes it very difficult to give clear advice for future model development.

2. For the benefit of the users of model outputs, I would like to see a discussion of where performance is 'good enough' for using the models for projections, and in what ways. I know this topic is difficult, but it is important context to judge the evaluation – at the moment, the reader is at a loss to know whether the old versions were not suitable for making particular climate projections but the new versions are suitable, or if both versions are still not useful, or both are good enough for a given purpose. This should be covered briefly throughout, in regard to each purpose, i.e. the simulation of mean rainfall to

projections of mean regional rainfall, and so on. There are some inferences made – e.g. page 21, line 6-10 suggests that models are not suitable to use for projections of changes to intra-seasonal variability in WAM rainfall in the old and the new version, and more like this would be useful.

The reviewer has a very good point. The usability of model results for supporting policy relevant decisions is an important aspect for users of the model data. The question of "how good is good enough" for using model results for any kind of application depends strongly on the process of interest. This includes details such as geographical region, simulated quantity, natural variability, time-scales and time range or metric (e.g. mean, extreme values, probability density function, etc.). We feel that statements on the usability or usefulness of the model results are beyond the scope of this study because of the mentioned complexity of the problem and also because of the high sensitivity of this topic. Similar to the model evaluation chapter (Flato et al., 2013) of the Fifth Assessment Report of the Intergovernmental Panel on Climate Change (IPCC-AR5), we regard "the need for climate models to represent the observed behaviour of past climate" only as a "necessary condition to be considered a viable tool for future projections". This does not answer the "much more difficult question of determining how well a model must agree with observations before projections made with it can be deemed reliable." (Flato et al. 2013). We added a brief general discussion of this topic to the introduction of the revised manuscript (p.2/l.24-31).

3. Throughout – for evaluation, why use just 20 years? I guess this is the IPCC baseline so is common, but it is short enough to be strongly affected by variability such as megadroughts. A longer period that still has satellite coverage (e.g. 1979-2016) would be better, then sub-periods within this could be also covered. If it is too expensive to run models for longer periods, then I understand this limitation is unavoidable and this should be mentioned. Also, figure captions need to note the time period in the figure captions (1986-2005).

The CMIP5 simulations analyzed (AMIP, historical) typically only cover the time period up to the year 2005. In order to maximize the comparability of the EMBRACE and CMIP5 simulations and with results from the model evaluation chapter of IPCC-AR5 (Flato et al., 2013), we decided to use the time period 1986-2005 as a common denominator even though sometimes more years are available. The figure captions have been updated to include the time periods shown (Figs. 3, 4, 5, 6, 7, 8, 9, 10, 15, 16, 17).

4. The bias plots nicely show the differences at high values, but I think bias plots are more useful if they include a middle section of blank/white where the bias is negligible rather than having colours go to zero – this helps interpreting areas where biases are small and avoid over interpreting differences between positive and negative when in fact they are not meaningful (goes for all figures)

The color scales are similar to the ones used in chapter 9 of IPCC-AR5 (Flato et al., 2013). We would therefore prefer to keep the color scales as they are as this allows for an easier comparison with the multi-model mean results shown in the IPCC-AR5. This has been clarified in the revised manuscript (p.5/l.22-24 and p.9/l.15-p.10/l.1).

5. Pg 26 line 8-16 – this is one school of thought about ENSO and the tropical Pacific, but others would disagree – it is important to cover a range of ideas here. E.g. work by Felicity Graham: Graham et al. 2014 Effectiveness of the Bjerknes stability index in representing ocean dynamics (Climate Dynamics), and Graham et al. 2015 Reassessing conceptual models of ENSO (Journal of Climate)

We added the alternative conceptual model for ENSO as suggested by the reviewer to the introduction of section 3.3 (p.42/l.19-22).

6. Specify the version of NCEP used in each case – NCEP1 or 2?

All NCEP data used are NCEP 1 data (Kalnay et al., 1996). This has been clarified in the revised version (Figs. 1, 4, 5, 9 and Tab. 3).

7. Pg 2, Line 15 – why IPCC 2007 not 2013?

We added IPCC 2013 (Flato et al., 2013) as a reference (p.2/l.17).

8. Fig 1 – I think an observations-based dataset should be used here, at least as the comparison panel, rather than two reanalyses. Also, are you reporting the difference between ERAint and NCEP as the observed uncertainty? If so, why not show panels only where the bias is larger than this observed uncertainty (blank out other regions)? Could do this in other figures too.

We are showing differences compared to ERA-Interim to allow for easier comparison with the CMIP5 multi-model mean shown in the IPCC-AR5 (Flato et al., 2013). The same is true for the color scale. We would therefore prefer to keep ERA-Interim as the reference dataset and the color scales as they are. The comparison with NCEP is shown to highlight the regions with particularly large uncertainties in the reanalyses. In order to strengthen this point, we added a panel showing the Climate Research Unit (CRU) dataset to the Fig. 1 and extended the discussion accordingly (p.6/l.27-30).

9. Section 3.1.2 – absolute errors in rainfall appear higher where the mean is higher (i.e. the tropics), so reduce the appearance of biases in drier areas. The paper needs a figure showing proportional bias (%), even in additional material, to give some perspective on rainfall biases outside the tropics – for example the biases in Canada, Australia and Siberia look small and almost indistinguishable in the different panels, but important differences could be seen in a % bias plot. If % biases are extremely large due to extremely low rainfall, then these areas could be masked out or else identified and discussed.

Following the suggestion of the reviewer, we added a figure showing the relative bias in precipitation to the supplementary material (Fig. S1).

10. Page 9, line 4 – I don't think projections will ever be 'accurate' in the sense they won't give a single, correct answer, so I think this word should be changed to "reliable", "robust" or similar – projections that give useful information but are not a single 'accurate' answer.

We replaced "accurate" by "reliable" as suggested.

11. Figures 14-15 – I was expecting to see SST and rainfall bias map plots for the coupled versions (to see the shape of the warm pool, the extent of the cold tongue bias, the shape of the double ITCZ etc.) - one can see the temperature bias in Figure 1 somewhat, but it is not very clear. Perhaps a Pacific SST and rainfall bias map here or in additional material?

Following the suggestion of the reviewer, we added SST and rainfall bias maps zoomed in over the Pacific to the supplementary material (Figs. S3, S4).

12. Page 38, line 20-24 – I think this conclusion is incomplete, yes the lack of improvement with finer resolution certainly indicates that there are problems at the coarse resolution and apparent good performance is probably related to compensating errors. But it also shows that these issues are not

solved by finer resolution – thus indicating that either some critical threshold of resolution has not been reached (perhaps we need resolution somewhere <14 km) or else some other non-resolution factor is involved (e.g. parameterisations are not working well enough and the improvement to the cloud scheme didn't fix the problem). See major comment 1, I think this type of discussion and conclusion is needed more generally

We agree with the reviewer that the results suggest that there are most likely also factors other than horizontal resolution involved. We extended the discussion in the revised manuscript to include this conclusion (p.42/l.7-9 and p.54/l.24-27).

Minor

1. Pg 2, Line 3 – ITCZ not defined on first usage in text

2. Pg 6, line 6 – Similar not similarly

3. Figure 2 – caption notes lower right panel is noted as data from CMIP, this should be CMAP

4. Figure 3 – the box with 'Reference' in it obscures an important high rainfall region in Nepal, suggest making it smaller and moving it (perhaps top left corner?), same in many other figures.

5. Page 15, line 7-8 – 1970s and 1980s don't need apostrophes,

6. Page 15, line 24 "that one of is the main" typo

7. Figure 10 caption and/or legend needs to explain the grey shading

8. Page 31, line 10 – a paper from 2010 can't show the 'current GCMs', note that this paper is about CMIP3

All minor suggestions / corrections have been applied as suggested.

[revised manuscript text omitted]